# A scalable generative model for dynamical system reconstruction from neuroimaging data

**Eric Volkmann**[1,2,*], **Alena Brändle**[1,3,4,*], **Daniel Durstewitz**[1,3,4,†], **Georgia Koppe**[3,5,6,†]

[1]Department of Theoretical Neuroscience, Central Institute of Mental Health (CIMH),
Medical Faculty Mannheim, Heidelberg University, Mannheim, Germany
[2]Institute for Machine Learning, Johannes Kepler University, Linz, Austria
[3]Interdisciplinary Center for Scientific Computing, Heidelberg University, Heidelberg, Germany
[4]Faculty of Physics and Astronomy, Heidelberg University, Heidelberg, Germany
[5]Hector Institute for AI in Psychiatry & Dept. for Psychiatry and Psychotherapy, CIMH
[6]Faculty of Mathematics and Computer Science, Heidelberg University, Heidelberg, Germany
[*,†] These authors contributed equally

## Abstract

Data-driven inference of the generative dynamics underlying a set of observed time series is of growing interest in machine learning and the natural sciences. In neuroscience, such methods promise to alleviate the need to handcraft models based on biophysical principles and allow to automatize the inference of inter-individual differences in brain dynamics. Recent breakthroughs in training techniques for state space models (SSMs) specifically geared toward dynamical systems (DS) reconstruction (DSR) enable to recover the underlying system including its geometrical (attractor) and long-term statistical invariants from even short time series. These techniques are based on control-theoretic ideas, like modern variants of teacher forcing (TF), to ensure stable loss gradient propagation while training. However, as it currently stands, these techniques are not directly applicable to data modalities where current observations depend on an entire history of previous states due to a signal's filtering properties, as common in neuroscience (and physiology more generally). Prominent examples are the blood oxygenation level dependent (BOLD) signal in functional magnetic resonance imaging (fMRI) or $Ca^{2+}$ imaging data. Such types of signals render the SSM's decoder model non-invertible, a requirement for previous TF-based methods. Here, exploiting the recent success of control techniques for training SSMs, we propose a novel algorithm that solves this problem and scales exceptionally well with model dimensionality and filter length. We demonstrate its efficiency in reconstructing dynamical systems, including their state space geometry and long-term temporal properties, from just short BOLD time series.

## 1 Introduction

Models and theories based on dynamical systems (DS) concepts have a long tradition in computational neuroscience in accounting for physiological phenomena and computational processes of the brain [74, 53, 33, 23]. Constructing such models from first principles (biophysics) is time-consuming and hard, and utilizing them to account for inter-individual differences in brain dynamics, when model settings need to be personalized, is even more challenging. Yet, constructing valid models of the brain's functional dynamics is immensely important, not only for understanding the neurocomputational basis of inter-individual differences in cognitive and emotional style [23], but also when aiming at

38th Conference on Neural Information Processing Systems (NeurIPS 2024).

diagnosing or predicting brain dysfunction and clinical characteristics based on DS features [66], or for designing personalized therapies [54].

## 1.1 Dynamical models in neuroscience

In computational neuroscience, multiple approaches to infer large-scale brain dynamics have been advanced over the past decades (e.g., [17, 46, 58, 32]). Pioneering work introduced latent linear DS, including Gaussian process latent variable models [64, 81], and extensions like Switching Linear DS (SLDS) [27]. Whereas linear DS models are very limited in their dynamical repertoire, SLDS offer a more flexible approach to modeling a larger range of dynamical phenomena by combining several linear (or affine) DS, with a switching mechanism that selects the active system at each moment [44, 43, 27]. These approaches have become common tools for inferring and visualizing neural trajectories within low-dimensional state spaces.

Arguably the most popular class of generative models in whole brain simulations relies on mean field neural population models, including neural mass [76, 5] and neural field models [35, 5]. These model the moments of the activity of cortical areas by averaging over properties (like firing rates) of neural (sub)populations, and are often biophysically motivated [17]. Many popular large-scale mean field modeling approaches are implemented in The Virtual Brain (TVB) environment [56, 58]. TVB incorporates biologically realistic brain connectivity into neural field models to generate simulations of large-scale brain activity. Alternatively, Dynamic Causal Modeling (DCM) describes a set of more statistically motivated and mostly linear techniques, primarily for the purpose of inferring the effective connectivity between brain regions based on invasive or non-invasive brain recordings (e.g., [16, 37, 46]). While many of these models may account for aspects of the dynamics, like patterns of functional connectivity and their task modulation, they are not, strictly, dynamical systems reconstruction (DSR) tools, entailing that they may miss important dynamical phenomena by being constrained through the biological assumptions and simplifications imposed.

In the analysis of functional magnetic resonance imaging (fMRI), only recently a shift in focus has introduced data-driven, deep-learning based methods for inferring generative models of system dynamics [40, 62, 63]. Models that implement dynamics either directly on the observed [62], or within an underlying latent [40, 63] space have been proposed, partly using available structural information and hierarchical inference approaches [63].

## 1.2 Dynamical systems reconstruction (DSR)

A variety of Deep Learning (DL)-based models for approximating the generative dynamical processes underlying observed time series has been put forward in recent years ([8, 69, 49, 40, 60, 42, 6, 71, 31, 13, 80]; see also [24] for an overview). These include approaches which approximate a system's vector field, e.g., through a library of basis functions and penalized regression as in SINDy [8], or through deep neural networks and neural Ordinary Differential Equations (ODE) [14, 57, 39]. Alternatively, methods that approximate the associated flow (solution) operator directly have been suggested, often employing state space model (SSM)-type architectures which distinguish between an observation process and a latent process commonly instantiated through recurrent neural networks (RNNs; e.g., [40, 29, 70, 60, 7]). Recurrent SLDS (rSLDS) [44], an extension of SLDS, and Latent Factor Analysis via Dynamical Systems (LFADS) [48] also fall into this category.

In DSR we ask for models that are *generative* in the sense that – after training – they provide an executable surrogate model of the observed system, from which we can simulate samples that agree with their empirical counterparts in topological, geometrical and temporal characteristics (in contrast to [24]. This required agreement in *long-term* temporal and geometric properties is not automatically guaranteed for standard training of common RNNs or neural ODEs, which may yield good short-term predictions but may fail to recover the full system dynamics [34, 24, 52]. Recent breakthroughs in data-driven DSR build on insights from the field of chaos control and synchronization [51, 1, 68, 2], by guiding the training process through optimally chosen control signals – modern variations of classical teacher forcing (TF) – that prevent exploding gradients [47, 6, 31, 7, 39].

Chaotic dynamics in particular, as typical for neural systems (e.g., [67, 22, 36, 26]), poses a severe problem here as trajectories and hence loss gradients inevitably diverge due to the presence of a positive Lyapunov exponent [47]. Recent amendments of TF protocols, including sparse TF

(STF; [47]) and generalized TF (GTF; [31]), keep trajectories and gradients in check by 'weakly synchronizing' them with the observed signals.

## 1.3 Specific contributions

Nonlinear SSMs distinguish between an underlying latent process that governs the dynamics of a system's state, and an observation process (referred to as *observation* or *decoder* model), that links the system states to the actual measurements [20]. Invertible (or pseudo-invertible) decoder models play a crucial role in control-theoretic approaches, like STF or GTF, for training SSMs, in order to project observations into the model's latent space. This inversion is fairly straightforward when ones assumes that the current measurement depends solely on the present latent state, i.e, when $x_t = f(z_t)$, but not if it depends on a history of states $\{z_t, z_{t-1}, \ldots, z_{t-\tau}\}$. In practice, unfortunately, this assumption is often violated due to a signal's filtering properties. For instance, blood oxygenation level dependent (BOLD) signals, as assessed via fMRI, are only an indirect measurement of neural activity, with the hemodynamic response function ($hrf$) broadly smearing out the signal across time. Each measurement is therefore a function of a history of past neural (latent) states whose dynamics we wish to infer [28, 78]. Similar challenges arise in calcium imaging when spike times are to be inferred [79, 72], or, in fact, any other observation process where the actual measurement is a filtering of the process of interest.

Here we rectify this issue by developing a particularly efficient SSM approach which works for measurements that depend on longer histories of latent states, yet allows to take advantage of recent powerful training strategies for DSR [47, 31]. In particular, our contributions are threefold: First, we create and validate a novel SSM-based DSR algorithm for observation models which involve convolutions with latent state series, and demonstrate its scalability with SSM size, as well as convolution filter length. Second, we introduce an evaluation scheme for selecting DSR models on short empirical time series, by demonstrating that the used DSR measures assessed on short time series accurately predict a system's long-term temporal and geometric properties. This is of high practical relevance, as in many empirically relevant scenarios, like fMRI, we only have access to comparatively short time series. Finally, we show that the proposed models can reliably extract key DS features that, moreover, differentiate between subjects.

## 2 Convolution SSM model (convSSM)

### 2.1 Latent DSR model

We start by defining our generative model used for DSR, a variant of a piecewise linear RNN (PLRNN). Its specific architecture has first been suggested in [21] and later expanded to increase the PLRNN's expressivity [6, 31]. While the present approach is generic and independent of the specific DSR architecture, here we use the so-called shallow PLRNN (shPLRNN; Appx. A.3) and the clipped shallow PLRNN (cshPLRNN; [31]). In the cshPLRNN, the temporal evolution of the (latent) system state $z_t \in \mathbb{R}^M$ is expressed as

$$z_t = \boldsymbol{A}z_{t-1} + \boldsymbol{W_1}\left[\phi\left(\boldsymbol{W_2}z_{t-1} + h_2\right) - \phi\left(\boldsymbol{W_2}z_{t-1}\right)\right] + h_1 \tag{1}$$

where $\phi(\cdot) = \max(0, \cdot)$ is an elementwise ReLU activation function, $\boldsymbol{W_1} \in \mathbb{R}^{M \times L}$, $\boldsymbol{W_2} \in \mathbb{R}^{L \times M}$ are connection weights, $\boldsymbol{A} \in \mathbb{R}^{M \times M}$ is a diagonal matrix of autoregressive weights, and $h_1 \in \mathbb{R}^M$ and $h_2 \in \mathbb{R}^L$ are bias vectors. Its trajectories $\{z_t\} \in \mathbb{R}^{M \times T}$ will be bounded if the absolute eigenvalues of $\boldsymbol{A}$ are smaller than $1$ [31]. The Markov property of the latent model is crucial to ensure it formally constitutes a DS with complete state space [50, 65]. Finally, Equation 1 can easily be extended to incorporate the effect of external inputs, such as experimental stimuli, by adding $\boldsymbol{C}s_t$ (with $s_t \in \mathbb{R}^K$ representing an input vector and $\boldsymbol{C} \in \mathbb{R}^{M \times K}$ its effect on the latent dynamics). However, here we consider input-free data from resting state experiments.

### 2.2 Teacher forcing for invertible decoder models

In [31], the latent state $z_t$ at each time point is assumed to be related to the actual observation $x_t \in \mathbb{R}^N$ by a linear (Gaussian) decoder model

$$x_t = \boldsymbol{B}z_t + \eta_t, \ \eta_t \sim \mathcal{N}(0, \boldsymbol{\Gamma}), \tag{2}$$

referred to simply as 'standard SSM' in the following. Here, $\boldsymbol{B} \in \mathbb{R}^{N \times M}$ is a matrix of regression weights, and $\eta_t$ describes Gaussian observation noise with diagonal covariance matrix $\boldsymbol{\Gamma}$. A conventional mean squared error (MSE) type loss function $\mathcal{L} = \sum_t \mathcal{L}_t = \sum_t \|\hat{x}_t - x_t\|_2^2$ between the generated (predicted) $\{\hat{x}_t\}$ and the observed $\{x_t\}$ sequence is commonly used to optimize parameters by stochastic gradient descent (SGD) with GTF [31]. Regularization terms to enforce a structure in latent space that helps to map slowly evolving processes may further be added to this loss [60].

A fundamental issue in training such systems by SGD is the well-known 'exploding-and-vanishing gradients' problem (EVGP), preventing systems from capturing relevant time scales in the data. In fact, [47] proved that for chaotic systems gradient-based training techniques for RNNs will *inevitably* lead to diverging loss gradients (see also [26]). Successful DSR algorithms need to address this problem. Based on this connection between chaos and diverging gradients, Engelken [25] suggested regularizing the system's Lyapunov spectrum, thereby also biasing the dynamics toward certain (non-hyperbolic) solutions. A theoretically well founded approach that does not limit a system's dynamical expressivity, which we will adopt here, is GTF, proposed in [31]. GTF is designed to keep model generated trajectories on track and, theoretically, can completely abolish the EVGP without constraining model expressivity. The main idea is that *during training* the latent state $\tilde{z}_t$ is computed as a linear interpolation between the PLRNN generated state $z_t = \text{PLRNN}(\tilde{z}_{t-1})$ and a data-inferred state $d_t$ that serves as a control signal [19], i.e.,

$$\tilde{z}_t := (1 - \alpha) \cdot z_t + \alpha \cdot d_t, \ \alpha \in [0, 1). \tag{3}$$

There is a theoretically optimal choice for $\alpha$ that can be approximated concurrently whilst training through a specifically designed annealing protocol [31], but more simply $\alpha$ may just be determined by grid search (as done here). Control signals $d_t$ are obtained by inverting the decoder model (Equation 2). Since in general $M \neq N$, the matrix inverse of $\boldsymbol{B} \in \mathbb{R}^{N \times M}$ does not exist and is approximated by the Moore-Penrose (pseudo-) inverse $\boldsymbol{B}^+$:

$$d_t = \boldsymbol{B}^+ x_t \tag{4}$$

To keep the gradients on track, the interpolation is performed at each time step before applying the cshPLRNN mapping (Equation 1). These control signals are turned off during actual data generation by the model (i.e., in a test phase), where it runs completely autonomously.

## 2.3 Teacher forcing for decoder models with signal convolution

**Decoder model for convolved signals** GTF (and similar techniques like STF; [47]) are powerful state-of-the-art (SOTA) tools for controlling gradients, especially in the context of DSR. However, they require a (pseudo-)invertible observation model for producing adequate control signals. Empirically, there are many situations where this requirement is not met. For instance, in BOLD time series the observed signal is a highly filtered and strongly smoothed version of the underlying neuronal process that we would like to recover [9, 10, 28]. This fairly complex hemodynamic process is often modeled by the $hrf$ [28].

A decoder model that relates the neuronal processes given as latent time series $\{z_t\}$ to measured BOLD time series $\{x_t\}$ may be formulated as in [40],

$$x_t = \boldsymbol{B} \left( (hrf * z)_t \right) + \boldsymbol{J} r_t + \eta_t, \ \eta_t \sim N(0, \boldsymbol{\Gamma}) \tag{5}$$

with regression coefficient matrices $\boldsymbol{B} \in \mathbb{R}^{N \times M}$ and $\boldsymbol{J} \in \mathbb{R}^{N \times P}$, nuisance variables $r_t \in \mathbb{R}^P$ (such as movement or respiratory artifacts) and a Gaussian observation noise term $\eta_t$ (with usually diagonal covariance $\boldsymbol{\Gamma} \in \mathbb{R}^{N \times N}$). Here, $*$ denotes the convolution operation and $z$ is a history of states $z_{t-\tau:t}$, the length of which depends on the observed sampling rate, commonly referred to as time of repetition (TR). The discrete $hrf$ sequence is computed by evaluating the canonical $hrf$ at the observed TR [78]. We will denote the $hrf$ response for a given TR by $hrf_{TR}$.

By incorporating the $hrf$ into the observation model, we disentangle the neural state and its dynamics – the processes of interest – from the neurovascular mechanics (or any filtering at the level of observed signals). We thereby eliminate the history dependence present in the observations, and thus help unfolding trajectories in latent space and satisfying the uniqueness assumptions required in reconstructing dynamical systems [50] (see also Appx. Figure 7). However, Equation 5 poses a major complication for applying TF techniques, as observations (and model outputs $\hat{x}_t$) do not simply depend on the current state $z_t$, but – due to the convolution – on a set of states across several previous time steps. We can thus not compute the control signal $d_t$ through straightforward decoder inversion anymore, but require a new type of inversion algorithm.

**Wiener deconvolution**  Following [78], we use a Wiener filter [75] to invert Equation 5. We briefly introduce this approach here in the context of our specific problem, and refer to Appx. A.4 for further details. Given an observed noisy signal $\{x_t\}$, composed of the signal of interest $\{z_t\}$ convolved with a known impulse response $hrf$ plus some noise term $\eta_t$ (distribution unknown, Wiener is optimal for Gaussian distribution),

$$x_t = (hrf * z)_t + \eta_t, \tag{6}$$

the Wiener deconvolution provides the estimate $\hat{z}_t$ of the unknown signal $z_t$ through least-MSE estimation. Defining $\text{Conv}^{-1}(\cdot, hrf)$ as the Wiener deconvolution operator, we can write

$$\{\hat{z}_t\} = \text{Conv}^{-1}(\{x_t\}, hrf). \tag{7}$$

**Inversion of BOLD decoder model**  Using the notation introduced above, we can write the inversion to obtain control signals as

$$\{d_t\} = \text{Conv}^{-1}\big(\{\boldsymbol{B}^+(x_t - \boldsymbol{J}r_t)\}, hrf\big), \tag{8}$$

where $\{\boldsymbol{B}^+(x_t - \boldsymbol{J}r_t)\}$ is the time series that needs to be deconvolved. Note that this approach is quite general and we can simply exchange the $hrf$ with alternative functions if we want to account for filtering in the original signal. As stated, since $\boldsymbol{B}$ and $\boldsymbol{J}$ are matrices of learnable parameters updated during training, we would need to perform this deconvolution step at every training epoch, which is computationally very costly. We therefore make use of the linearity of convolutions and separate the deconvolution step from the learnable parameters, rewriting Equation 8 as

$$\{d_t\} = \boldsymbol{B}^+\big(\text{Conv}^{-1}(\{x_t\}, hrf) - \boldsymbol{J} \cdot \text{Conv}^{-1}(\{r_t\}, hrf)\big). \tag{9}$$

With $\{x_t^{\text{deconv}}\}$ and $\{r_t^{\text{deconv}}\}$ denoting the respective deconvolved time series, this can be written as

$$d_t = \boldsymbol{B}^+\big(x_t^{\text{deconv}} - \boldsymbol{J} \cdot r_t^{\text{deconv}}\big). \tag{10}$$

This now is computationally much more efficient, as we need to perform the deconvolution only once prior to training. During training then, only the decoder model parameters need to be inferred to obtain the control signal. We will refer to the convolutional model for DSR (Equation 1 and Equation 5) trained with GTF and SGD as 'convSSM'. The full inversion algorithm is provided in Algorithm 1 with additional information given in Appx. A.6. Key components of SGD+GTF training are illustrated in Figure 1.

## 3  Results

### 3.1  Performance measures

In DS theory in general, and DSR more specifically, we are mostly concerned with invariant properties of a system, such as attractor geometry and long-term temporal statistics [24]. In chaotic systems in particular, in which trajectories diverge exponentially fast with time, the mean squared prediction error (PE) is a useful statistic only on relatively short time scales [77, 40, 60] (see Appx. Figure 6). To evaluate our model's performance, in addition to short-term $n$-step ahead PEs, $PE_n$, we assess the following two established performance measures to capture the temporal and geometrical structure:

1. The deviation in power spectra between the (smoothed) empirical and model-generated power spectra, assessed in terms of the Hellinger distance and referred to as power spectrum error (PSE), $D_{PSE}$, in the following [47], and

2. the Kullback Leibler divergence between the empirical and model-generated trajectories across state space, $D_{stsp}$, measuring the overlap in attractor geometries [40].

To obtain a reference value for $D_{stsp}$, we further included two references in which we assessed $D_{stsp}$ when all mass is centered on the expectation value (similar to a fixed point solution), and when the state space is populated by points drawn from a Gaussian with mean and variance equal to the data (similar to a fixed point solution plus measurement noise).

For comparability with experimental data, we evaluated performance on comparatively short time series obtained from the adaptive linear-nonlinear (ALN) cascade model and the LEMON data set. In these cases, performance metrics were assessed on 100 generated trajectories per model and then averaged. For more details, see Appx. A.7.

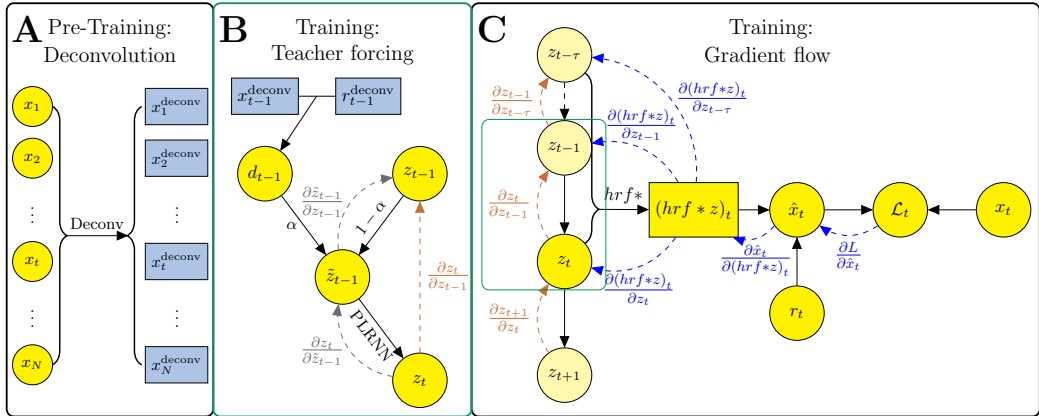

Figure 1: Schematic of training protocol and gradient flow. A: Before training, observations $\{x_t\}$ and nuisance artifacts $\{r_t\}$ are deconvolved. B: The deconvolved time series are used to generate a forcing signal $d_{t-1}$ which is used for guiding cshPLRNN training. C: Latent states $z_{t-\tau:t}$ and nuisance artifacts $r_t$ are used to predict $\hat{x}_t$ through the decoder model. Gradients are computed on the squared error loss $\mathcal{L}_t$, propagated from the decoder model back to the latent states (blue), and from the latent DS model backwards in time (orange).

### 3.2 convSSM validation & scalability on Lorenz63

As a well-established and popular benchmark for a chaotic system, we first performed numerical experiments with the famous Lorenz63 system. The Lorenz63 is a 3-dimensional system introduced in [45] to describe atmospheric convection, and exhibits chaotic behaviour in the chosen regime (see Appx. B.1). To mimic BOLD observations, we generated 100 standardized chaotic Lorenz63 trajectories, convolved them with $hrf_{TR}$ functions at different sampling rates TR $\in \{0.2\text{s}, 0.5\text{s}, 1.2\text{s}\}$ (Figure 2B), and added Gaussian noise with standard deviation $\sigma \in \{0.01, 0.1\}$. This resulted in 6 benchmark settings with different levels of signal degradation by convolution and noise. Each data set was divided into a training and a test set of $T = 5 \cdot 10^4$ time steps each.

We trained 100 models on each of these 6 data sets. The following models were compared: the convSSM trained via SGD+GTF, the convSSM trained via SGD and no GTF, the standard SSM trained via SGD+GTF, and MINDy, a recently published method for DSR in fMRI [62]. convSSM and standard SSMs were trained with the shPLRNN, with $M = 3$, $L = 50$, and $\alpha = .1$ (see Appx. Table 4 for all additional hyperparameters). The hidden dimension was selected such that the standard SSM (no-$hrf$ model) performed well [31]. We emphasize that the standard SSM has already been extensively benchmarked on a variety of simulated and real-world data sets and is considered to be a SOTA model in the field [6, 31]. The performance measures $D_{stsp}$, $D_{PSE}$, and $PE_{20}$ were assessed on the test sets after training for $1,000$ epochs. We used the same hyperparameters for all networks (aside from TR) to show that performance increases can be solely attributed to the improved decoder model. Hyperparameters were chosen such that the shPLRNN achieved near perfect performance on *unconvolved*, noiseless trajectories from the Lorenz63 system.

The convSSM significantly outperformed all other methods, including the standard SSM in almost all cases, with the performance gap increasing with decreasing TR (see Appx. Table 2 for performance, and Figure 2A for example reconstructions, providing an intuition on how to interpret $D_{stsp}$). The more heavily the signal was degraded by the convolution filter, the larger was the performance gap in favor of the convSSM.

An important consideration especially for large-scale applications of such models to empirical data is how well they scale with model size and convolution filter length. To assess this, we collected trajectories of length $T = 10^5$ from the chaotic Lorenz63 system, and studied training epoch times as a function of convSSM latent dimension $M = \{3, 10, 50, 100, 500\}$, hidden dimension $L = \{10, 50, 100, 500, 1000\}$, TR $= \{0.2, 0.5, 1.2, 3\}$, time series length $T = \{500, 1000, 5000, 10000, 50000, 100000\}$ and observation dimension

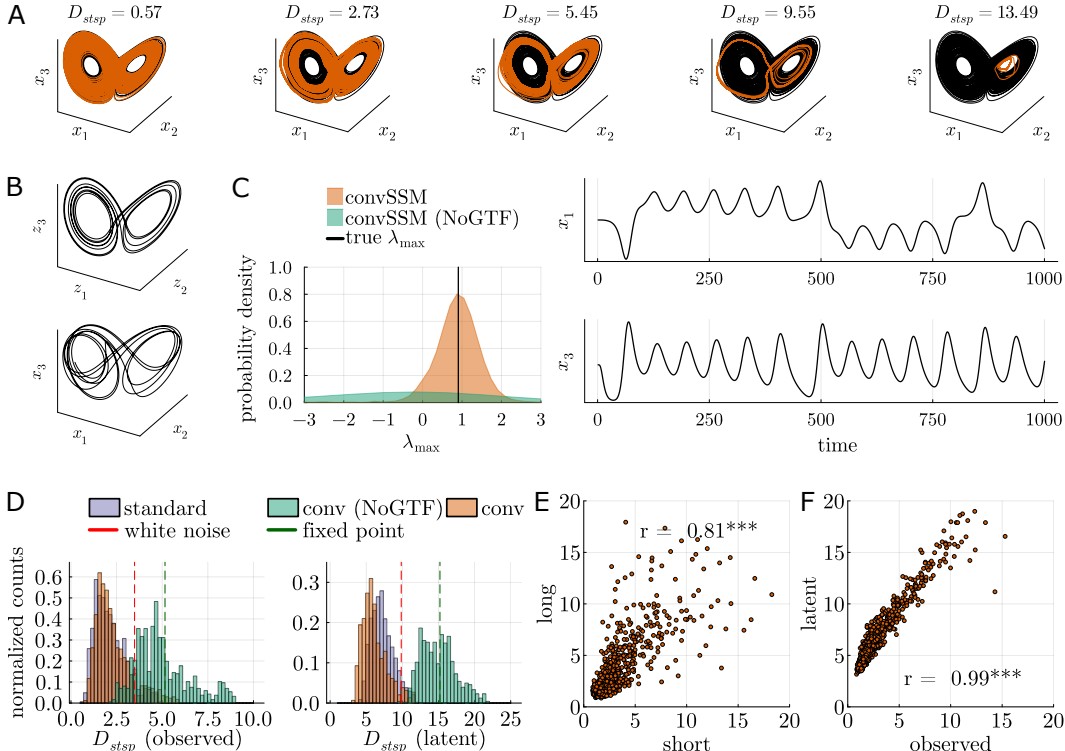

Figure 2: Validations on Lorenz63 and ALN. A: Illustration of reconstruction performance as assessed by the geometrical agreement measure $D_{stsp}$. Average $D_{stsp}$ values for the convSSM were $D_{stsp} < 0.30$ at noise level $\sigma = .01$ and $D_{stsp} < 0.71$ at noise level $\sigma = .1$, indicating successful reconstructions in the majority of cases. B: Example trajectory from the Lorenz63 system in latent space (top) and observation space (convolved with $hrf_{0.2}$) (bottom). C: Probability density over maximal $\lambda_{max}$ values (orange) assessed on 1000 convSSMs trained on Lorenz63 time series of length 1000 (example shown in right panel). Black line denotes the known $\lambda_{max} \approx 0.9056$ of the Lorenz system. D: Comparison of standard SSM ('standard'), convSSM ('conv'), and convSSM trained without generalized teacher forcing ('conv (NoGTF)') on the ALN data set. Histograms over $D_{stsp}$ assessed on the observed space (left panel) and latent space (right panel). E: $D_{stsp}$ for convSSM evaluated on the full pseudo-empirical time series of typical empirically available length ($T = 500$; *x-axis*) vs. the long GT test set ($T = 5,000$; *y-axis*). F: $D_{stsp}$ for convSSM evaluated on the observed time series (*x-axis*) vs. on the latent time series (*y-axis*).

$N = \{10, 30, 50, 100, 500, 1000\}$. Shorter/longer TRs directly implicate longer/shorter convolution filters since the filters assume a constant time interval. Results are displayed in Appx. Figure 4A. The runtime per epoch did not significantly depend on TR, which means that time series convolved with long impulse response functions can be trained in times comparable to short ones. The per-epoch-runtime increases approximately linearly with dimensions $L$, $M$, and $N$ (Appx. Figure 4 B,C,E), implying that models can be scaled up efficiently.

Finally, empirical data is often short, yet we want to reliably infer DS features that characterize the underlying dynamics. To demonstrate that our model can robustly reconstruct dynamics based on short time series, we inferred 1000 convSSMs on $n = 100$ convolved Lorenz63 time series (TR $= 0.5$) of length $T = 1000$ only (see Figure 2C). We then assessed the degree of chaoticity in the recovered trajectories by examining the trained models' maximum Lyapunov exponents, $\lambda_{max}$. $\lambda_{max}$ measures how quickly trajectories starting from nearby points in a system's state space converge or diverge with time. If $\lambda_{max} > 0$, trajectories will exponentially diverge and the system, if bounded, will exhibit chaos. We show that we can successfully recover $\lambda_{max}$ (known for the Lorenz63 system; Figure 2C) even from models trained on these just short series.

### 3.3 Validating performance measures on short time series

In empirical situations, we do not have access to the latent dynamics of the true system, of course, but we still rely on our reconstruction measures evaluated on the *observed signals* to yield results valid for the (unobserved) latent space. It is therefore a practically very relevant question whether a) the convSSM trained on such short time series would be able to accurately recover the underlying neural latent dynamics, and b) our measures $(D_{stsp}, D_{PSE})$ evaluated on such short time series, and directly on the observations, would yield results similar to what would be expected if much longer time series and access to the ground truth latent space were available.

To tackle these questions, we used a more realistic simulation model, the ALN model [3, 11] for simulating whole brain (neural) activity. 100 data sets of length $T = 10,000$ were simulated from this model using *neurolib* [12], sampled at $0.1\,\mathrm{ms}$, and filtered through Equation 5 (with TR = $0.1\,\mathrm{ms}$) to compute the corresponding BOLD time series. Subsequently, these time series were downsampled to a TR of $0.5\,\mathrm{s}$ to mimic an experimentally realistic scenario (see Appx. B.2 for details). Most hyperparameters were adopted from previous experiments (see Appx. Table 4). For the latent dimension, we chose $M = 16$ to match the dimensions of the empirical LEMON data set, see subsection 3.4.

To mimic real fMRI experiments, we then pretended that only the first 500 time points are available for model estimation (called 'pseudo-empirical' here to distinguish it from the actual empirical LEMON data set). We trained 10 convSSM models on the first 375 time steps of each of these virtual experiments, treating the left out 125 time points as pseudo-empirical test set and call the last $5,000$ time points of the entire trajectory (i.e., time steps $5,001$-$10,000$ of the full simulation set) the ground truth (GT) test set (which would not be accessible in a real experiment). DSR performance was assessed on both the observed $\{x_t\}$ and latent $\{z_t\}$ time series, evaluated for a) the short pseudo-empirical test set of length 125, b) the full pseudo-empirical time series (i.e., of length 500), and c) the GT test set of length 5000 (which also assesses dynamics on the limit set and does not contain transients anymore).

Figure 2D shows histograms over $D_{stsp}$ (on the full pseudo-empirical time series) for the convSSM, the standard SSM, and the benchmark conditions. The convSSM significantly outperformed the standard SSM in latent space (rank-sum test $Z = 11.50$, $p \leq .001$), demonstrating an improved recovery of the ground truth DS and indicating that the deconvolution acts as an inductive bias that forces the model to learn a latent space structured in agreement with our biophysical understanding of fMRI. Moreover, the proposed performance measures $(D_{stsp}, D_{PSE})$ successfully discriminated between good and poor reconstructions even on these short time series more typical for empirical data: For one, evaluating DSR on the observations was consistent with evaluating DSR directly on the latent dynamics space (Figure 2F and Appx. Figure 5). Second, DSR assessed on the pseudo-empirical time series (either full or only test set) was strongly correlated with performance assessed on the long GT test set (which, again, in empirical situations we do not have, Figure 2E and Appx. Figure 5).

### 3.4 Application to experimental fMRI data

We finally tested convSSM on empirical data, for which we chose the LEMON study ('Leipzig Study for Mind-Body-Emotion Interactions') as a publicly available data set. This data set was collected at the Max-Planck-Institute Leipzig [4] and consists of 227 healthy participants, each of whom completed a battery of tests, including a 15min 30s resting state fMRI (rsfMRI) session sampled at TR = $1.4s$ (thus comprising $T = 652$ time points). We used the preprocessed rsfMRI data sets as provided, and selected 16 regions from which we extracted a subset of the available time series. These were subsequently smoothed, band-pass filtered, and standardized as in [40]. The time series were split $3:1$ into training ($T_{train} = 489$) and test ($T_{test} = 163$) set, respectively. Data from participants with non-stable variance were discarded (i.e., non-stationary data, see Appx. B.3 for details), leaving $N = 51$ participants for analysis.

We trained 20 models on data of each participant with latent dimension $M = N = 16$ (i.e., equal to the observation dimension), $\alpha = .1$, and hidden dimension $L = 50$ (where $L$ and $M$ refer to the dimensions of the connectivity matrices $\boldsymbol{W_1}, \boldsymbol{W_2}$ in the cshPLRNN, Equation 1). Latent dimension and $\alpha$ were determined via grid search, by inferring systems using a subset of the data and assessing the performance on the held-out set [6]. Otherwise the same hyperparameters as used in [31] for EEG data were applied (see Appx. Table 4 for all details). In addition to the model comparisons

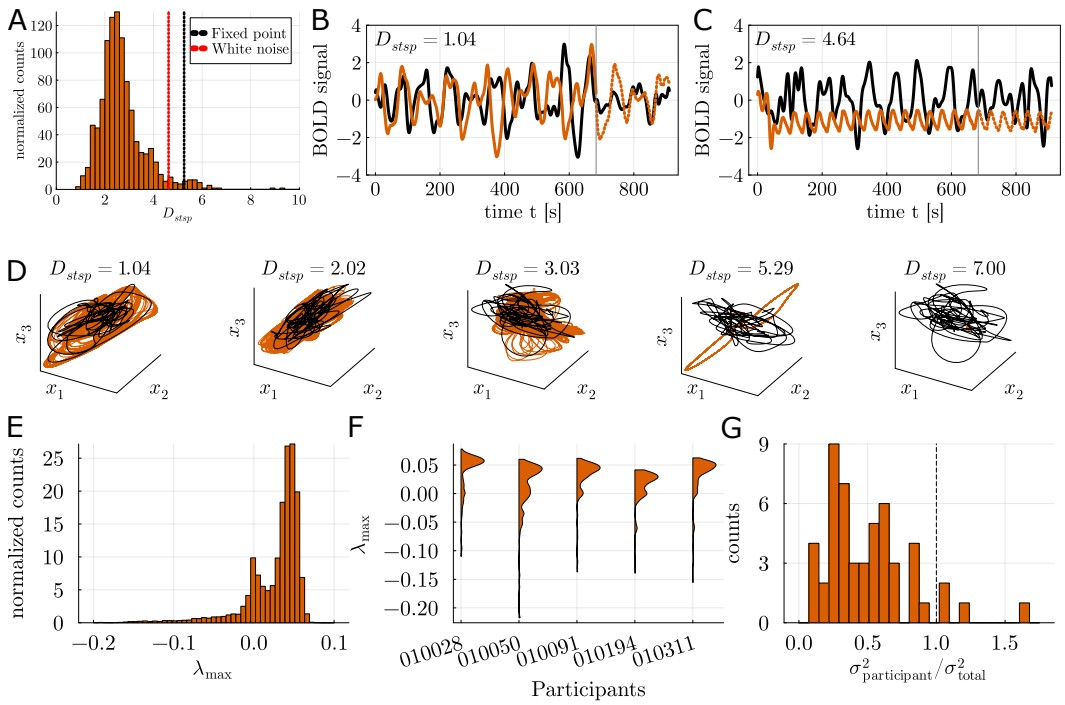

Figure 3: Results on empirical LEMON data set. A: Distribution over $D_{stsp}$ for 1020 systems inferred with convSSM. B: Example of a good and C: poor reconstruction. D: Illustration of reconstruction performance as a function of $D_{stsp}$. E: Histogram over maximum Lyapunov exponents $\lambda_{max}$. F: Distribution over $\lambda_{max}$ for 5 selected participants ($n = 100$ systems with 10 trajectories each). G: Within- as compared to between-subject variance in $\lambda_{max}$ distribution after filtering models by DS performance measures (selecting the 20 best by $D_{stsp}$ and 10 best by $D_{PSE}$).

discussed earlier, we also compared our method to the performance of rSLDS [44] and LFADS [48] (see Appx. C.2 for details). On top of $D_{ststp}$, $D_{PSE}$, and $PE_n$, we also assessed the trained models' maximum Lyapunov exponents, $\lambda_{max}$, analyzed how reliably these can be inferred, and whether they distinguish between subjects. Note that obtaining an estimate of the Lyapunov exponent is an advantage of the generative model, as empirical time series are often too short to compute it reliably. Also, since we do have access to the cshPLRNN's Jacobians, the computations can be performed analytically (although practically we need to evaluate these along model-generated trajectories, where here we used an algorithm proposed in [73]).

The DSR results are shown in Table 1. We obtained successful reconstructions on average with a mean $D_{ststp}$ of 2.73, better than all other models (Figure 3A and D). Interestingly, most recovered systems were characterized by a positive maximal Lyapunov exponent $\lambda_{max}$ (Figure 3E), indicating the presence of chaotic attractors in these data (consistent with previous observations, [40, 36, 41]). Moreover, $\lambda_{max}$ values could be inferred reliably (Figure 3F), and differentiated between individuals, as indicated by lower within- as compared to between-subject variation (Figure 3G, $T(50) = -11.53, p < .001$.

Table 1: DSR measures evaluated for the convSSM, standard SSM, convSSM trained without GTF, as well as MINDy [62], rSLDS [44] and LFADS [14], trained on the LEMON dataset. Model runs were excluded if the 1-step PE > 1 on the training data.

| metric | ConvSSM | standard SSM | No GTF ($\alpha = 0$) | MINDy | rSLDS | LFADS | Noise process | Fixed point |
|---|---|---|---|---|---|---|---|---|
| $D_{stsp}$ | $2.73 \pm 1.09$ | $2.77 \pm 0.93$ | $3.77 \pm 1.22$ | $6.79 \pm 1.92$ | $15.51 \pm 10.02$ | $3.24 \pm 1.17$ | $4.62 \pm 0.91$ | $5.27 \pm 1.27$ |
| $D_{PSE}$ | $0.14 \pm 0.03$ | $0.15 \pm 0.03$ | $0.34 \pm 0.11$ | $0.27 \pm 0.06$ | $0.24 \pm 0.03$ | $0.43 \pm 0.09$ | $0.76 \pm 0.02$ | - |
| 10-step PE | $1.78 \pm 0.38$ | $2.00 \pm 0.44$ | $1.43 \pm 0.61$ | $1.97 \pm 0.31$ | $1.78 \pm 0.42$ | $2.45 \pm 0.55$ | - | - |

# 4    Conclusions

Methods for producing generative models of the underlying dynamics from time series observations is a rapidly expanding research field [24]. Current SOTA models for this purpose rely on control theoretically motivated training techniques like STF [47] and GTF [31], but these require some means to generate from the actual observations a TF signal in the model's latent space for guiding trajectory and gradient flows. This becomes difficult if the current observation depends on a whole series of latent states, as common if the actual measurements are some filtering of an underlying process of interest, such as in fMRI or $Ca^{2+}$ imaging. Here we provide a novel technique that efficiently deals with this problem, exploiting linearity of Wiener deconvolution. A hallmark of our technique is that it efficiently scales with model size and convolution length.

Another major contribution of this work is to numerically demonstrate that the short time series obtained in typical fMRI experiments are actually sufficient for proper model selection according to established DSR performance measures, and that these can indeed be properly evaluated in observation space and do not require access to the unobserved dynamics/ latent space. This is of major empirical relevance for many scientific scenarios, beyond fMRI, in which time series sampling is costly or restricted for technical reasons. Finally, using our DSR technique, we showed that experimental fMRI signals mostly exhibit properties of chaotic oscillators (consistent with [36]), and that these can be reliably inferred and differ between subjects. Taken together, these contributions pave the way for deploying data-driven fMRI DSR models at large scale to understand inter-individual differences in brain dynamics and explore the predictive value of nonlinear DS features for cognitive or clinical assessment.

We emphasize that the proposed framework is highly flexible due to its modular structure, and may be easily adapted to meet diverse requirements. First, the latent model can be replaced with any other differentiable and recursive dynamical model, such as e.g. LSTMs [59]. The GTF training framework would remain unchanged as the control signal and the latent state update (Equation 3) are not affected by such modifications [31]. Likewise, the observation model can easily be adapted to account for nonlinear effects of nuisance covariates, e.g. through basis expansions in these variables, or through learnable but regularized MLPs. While our model was designed as a scalable method to integrate biological prior knowledge on convolution filters like the $hrf$, alternatively we can parameterize the filter weights within the observation model, making them learnable through BPTT, with filter length either as a hyperparameter, or by imposing a regularization that truncates filter length by driving coefficients to zero. To prevent conflicts between filter adjustment and latent model, a viable strategy may be stage-wise learning as suggested in [40]. Once the filter is adjusted, one may reduce the learning rate on the observation model, or even fix its parameters, to prioritize learning of the dynamics. Fixing the filter parameters after an initial stage would have the advantage that subsequent training would enjoy the same speed benefits as in our suggested method.

We furthermore highlight that our framework could be adapted to accommodate noise in the latent process. For example, in Brenner et al. [7] the GTF procedure has been modified to work in the context of stochastic DSR models using variational inference. Instead of the multimodal encoder model in Brenner et al. [7], one may use the inversion in Equation 9 to generate a TF signal which steers a probabilistic latent DS model, i.e. controls its distributional mean, via Equation 3, and using the reparameterization trick [55, 38] for BPTT in latent space. However, although probabilistic frameworks are appealing, 'deterministic' BPTT has previously been shown to be (at least) comparable in terms of DSR performance, even for clearly noisy observations and latent processes [6], such that the benefits for DSR would need to be further examined.

**Limitations** Data-driven approaches such as the one proposed here lack detailed biophysical mechanisms and may thus not be as suited to address specific questions relating to pharmacological or receptor mechanisms beyond functional-dynamical implications. Moreover, currently open questions are how to best deal with non-stationarity in the data, how to efficiently combine data from many subjects, and how trained models generalize to out-of-domain data.

## Software and Data

Code for the convSSM is available at `https://github.com/humml-lab/GTF-ConvSSM`.

## 5 Acknowledgements

This work was supported by the German Research Foundation (DFG) within the collaborative research center TRR 265, subproject B08, granted to GK, TRR 265 subproject A06 granted to DD and GK, Germany's Excellence Strategy EXC 2181/1 – 390900948 (STRUCTURES), and the Hector II foundation.

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

# A  Further methodological details

## A.1  Deconvolution algorithm in convSSM

---

**Algorithm 1** Deconvolution in convSSM

---

**Input:**
   $\mathcal{X}$: measured time series as $(N \times T)$ matrix
**Parameters:**
   $\psi$: analyzing wavelet
   $\tilde{\sigma}_{min}$: minimum noise level
   $cut_l$, $cut_r$: edge cutoffs
   $hrf_t$: kernel of hemodynamic response function
**Output:**
   $\mathcal{X}_{deconv}$: deconvolved time series as $(N \times T)$ matrix

**Initialize** $\mathcal{X}_{deconv} := \text{zeros}(N, T)$
**for** $i = 1$ **to** $N$ **do**
   $x_t := \mathcal{X}[i, :]$
   $\tilde{\sigma}, \tilde{z}_t := \text{VISUSHRINK}(x_t, \psi)$
   **if** $\tilde{\sigma} < \tilde{\sigma}_{min}$ **then**
      $\tilde{\sigma} := \tilde{\sigma}_{min}$
   **end if**

   *Compute Fourier transforms $\mathcal{F}\{\cdot\}$ and expectation values of spectral densities $\mathbb{E}\left[|\mathcal{F}\{\cdot\}|^2\right]$*
   $X_k := \mathcal{F}\{x_t\}$
   $HRF_k := \mathcal{F}\{hrf_t\}$
   $N_k := \mathbb{E}\left[|\mathcal{F}\{\eta_t\}|^2\right]$
   $S_k := \mathbb{E}\left[|\mathcal{F}\{\tilde{z}_t\}|^2\right]$

   *Compute and apply Wiener filter*
   $W_k := \frac{HRF_k^* \cdot S_k}{|HRF_k|^2 \cdot S_k + N_k}$
   $\tilde{Z}_k := W_k \cdot X_k$

   *Transform back to time domain*
   $\tilde{x}_t := \mathcal{F}^{-1}\left\{\tilde{Z}_k\right\}$

   *Remove signal edges*
   $\tilde{x}_t[\text{begin} + cut_l : \text{end}] := \text{NaN}$
   $\tilde{x}_t[\text{begin} : \text{end} - cut_r] := \text{NaN}$
   $\mathcal{X}_{deconv}[i, :] := \tilde{x}_t$
**end for**

---

## A.2 Scalability

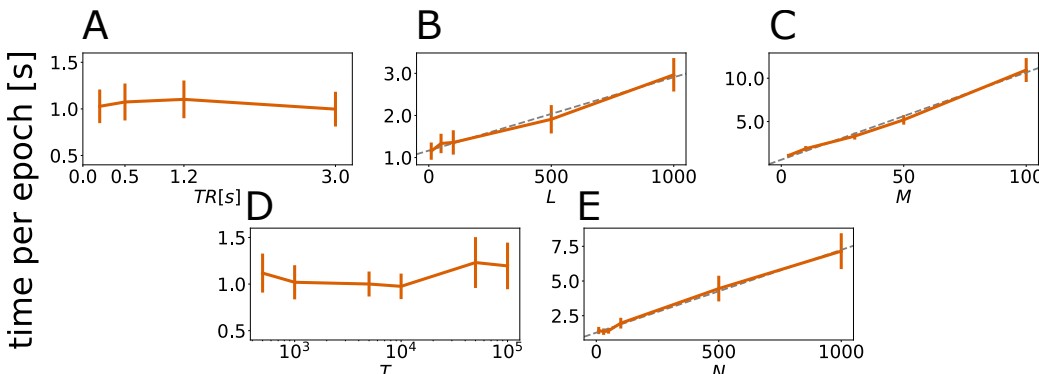

Figure 4: Training duration per epoch (y-axis) in seconds for different TRs (A), hidden dimensions L (B), latent dimensions M (C), time series length (D), and observation dimensions (E). Mean, standard error (SEM) and linear curve fits (gray dashed lines) are displayed. The per-epoch-runtime increases approximately linearly with dimensions $L$, $M$, and $N$; explained variance $R_L^2 = 0.989$, $R_M^2 = 0.993$, and $R_N^2 = 0.996$ for linear regressions with predictors '$L$', '$M$', and '$N$', respectively. Experiments were performed on a standard notebook with Intel i5-8250U 1,60 GHz CPU and 8GB RAM.

## A.3 PLRNNs

The simplest form of the PLRNN is given by

$$z_t = \boldsymbol{A} z_{t-1} + \boldsymbol{W} \Phi(z_{t-1}) + h. \tag{11}$$

where $z_t \in \mathbb{R}^M$ is a latent state vector at time $t$, $\Phi(\cdot) = \max(0, \cdot)$ is the ReLU activation function, $\boldsymbol{W} \in \mathbb{R}^{M \times M}$ is an off-diagonal matrix of connection weights and $\boldsymbol{A} \in \mathbb{R}^{M \times M}$ a diagonal matrix containing the autoregressive weights [21].

Neurobiologically motivated, the entries of the latent state $z_{i,t}$ may be interpreted as membrane potentials, the diagonal elements in $\boldsymbol{A}$ as the neurons' individual membrane time constants, and the off-diagonal elements in $\boldsymbol{W}$ as synaptic connections between neurons. The ReLU activation emulates that neurons only start spiking above a certain firing threshold.

By adding one hidden layer, we obtain the so-called shallow PLRNN

$$z_t = \boldsymbol{A} z_{t-1} + \boldsymbol{W_1} \phi \left(\boldsymbol{W_2} z_{t-1} + h_2\right) + h_1 \tag{12}$$

with $z_t \in \mathbb{R}^M$ latent states at time $t$, $\phi(\cdot) = \text{ReLU}(\cdot)$, diagonal matrix $\boldsymbol{A} \in \mathbb{R}^{M \times M}$, rectangular connectivity matrices $\boldsymbol{W_1} \in \mathbb{R}^{M \times L}$ and $\boldsymbol{W_2} \in \mathbb{R}^{L \times M}$ and thresholds $h_1 \in \mathbb{R}^M$ and $h_2 \in \mathbb{R}^L$.

## A.4 Wiener filter

The Wiener deconvolution filter is typically described in the frequency domain, and, for the setting in Equation 6 and Equation 7, is given by

$$W_k = \frac{HRF_k^* S_k}{|HRF_k|^2 S_k + N_k} \tag{13}$$

and returns the estimate $\hat{z}_t$ as

$$\hat{z}_t = \mathscr{F}^{-1}(W_k X_k) = \left(\text{Conv}^{-1}(\{x_{t'}\}, hrf)\right)_t \tag{14}$$

- where time series denoted with capital letters correspond to the Fourier transformation of time series denoted by lowercase letters, i.e., $\{X_k\} = \mathscr{F}(\{x_t\})$,

- $W_k$ is the Wiener filter,

- $S_k = \mathbb{E}[|Z_k|^2]$ is the mean power spectral density of the original signal $z_t$,

- $N_k = \mathbb{E}[|H_k|^2]$ is the mean power spectral density of the noise $\eta_t$,
- the superscript $^*$ denotes complex conjugation, and
- $\mathscr{F}^{-1}$ is the inverse Fourier transform.

The noise spectrum $N_k$ is typically unknown in practice, but can be reliably estimated based on the median estimator on the finest scale wavelet coefficients of $x_t$. As approximation to the power spectrum of the original signal, we use the denoised signal $\tilde{x}_t$ which we obtain by applying the VISUSHRINK algorithm [18], Algorithm 2, to the observed signal $x_t$. This approximation works well in practice in absence of knowledge about the true underlying signal.

---

**Algorithm 2** VISUSHRINK algorithm
___
**Input:** time series data $\{x_t\}$ of length $N$; analyzing wavelet $\psi$

1. Apply the discrete wavelet transformation (DWT) to the input data $\{x_t\}$ to obtain the wavelet coefficients $\Theta_t^0$

$$\Theta_t = \mathscr{W}_\psi\{x_t\} \tag{15}$$

2. Calculate the MAD of the fine scale coefficients $\Theta_t^0$, the estimate of the noise level $\tilde{\sigma}$ and the universal estimator $\lambda_U$. Let $\bar{\Theta} = \text{median}(\Theta_t^0)$ be the median of the fine scale coefficients and $N$ the length of the time series $\{x_t\}$

$$\text{MAD} = \text{median}(|\Theta_t^0 - \bar{\Theta}|) \tag{16}$$

$$\tilde{\sigma} = \frac{\text{MAD}}{0.6745} \tag{17}$$

$$\lambda_U = (2\ln N)^{1/2}\tilde{\sigma} \tag{18}$$

3. Apply hard thresholding to the fine scale wavelet coefficients

$$\Theta_t^1 = \begin{cases} 0, & |\Theta_t^0| < \lambda_U \\ \Theta_t^0, & |\Theta_i^0| \geq \lambda_U \end{cases} \tag{19}$$

4. Apply the inverse DWT to the tresholded coefficients $\Theta_t^1$ to obtain an estimate of the denoised signal $\{\tilde{x}_t\}$

$$\{\tilde{x}_t\} = \mathscr{W}_\psi^{-1}\{\Theta_t^1\} \tag{20}$$

**Output:** estimate of the noise level $\tilde{\sigma}$; estimate of denoised signal $\{\tilde{x}_t\}$

---

### A.5 Additional details on BOLD observation model

The discrete convolution of two time series $\{f_t\}$ and $\{g_t\}$ in its general definition is given by

$$(f * g)_t = \sum_{s=-\infty}^{\infty} f_s \cdot g_{t-s} \tag{21}$$

which in case of the decoder model given in Equation 5 translates to

$$(hrf * z)_t = \sum_{s=0}^{\tau} hrf_s \cdot z_{t-s} \tag{22}$$

due to the finite non-zero value of $\tau$, i.e., the maximum time difference at which a previous state $z_{t-\tau}$ still influences the current state $z_t$ according to the $hrf$. Furthermore, due to causality, only past states are allowed to influence the current state.

Since the convolution operation as well as matrix multiplication are linear, the order of convolution and matrix multiplication in Equation 5 can be inverted to yield Equation 9. This can be seen more easily if we consider a decoder model without noise

$$x_t = \boldsymbol{B}(hrf * z)_t. \tag{23}$$

Due to the $hrf$ being the same for all latent dimensions (i.e., a scalar and not a function), we interchange the matrix multiplication with $\boldsymbol{B}$ and the convolution with the $hrf$

$$\boldsymbol{B}(hrf * z)_t = \boldsymbol{B}\sum_{s=0}^{\tau} hrf_s \cdot z_{t-s} \tag{24}$$

$$(\boldsymbol{B}(hrf * z)_t)_i = \sum_{j=1}^{M}\boldsymbol{B}_{ij}\sum_{s=0}^{\tau}hrf_s \cdot z_{t-s,j} = \sum_{s=0}^{\tau}hrf_s\sum_{j=1}^{M}\boldsymbol{B}_{ij}\cdot z_{t-s,j} = \sum_{s=0}^{\tau}hrf_s(\boldsymbol{B}\cdot z_{t-s})_i \tag{25}$$

$$\boldsymbol{B}(hrf * z)_t = (hrf * (\boldsymbol{B}z))_t. \tag{26}$$

Consequentially, we divide the observations into two parts

$$x_t = \left(hrf * x^{\text{deconv}}\right)_t \Leftrightarrow \{x_t\} = \text{Conv}\left(\left\{x_{t'}^{\text{deconv}}\right\}, hrf\right) \tag{27}$$

$$x_t^{\text{deconv}} = \boldsymbol{B}z_t \tag{28}$$

The second part, Equation 28, is of the same form as Equation 2 and permits the same inversion. Therefore, by computing $\left\{x_t^{\text{deconv}}\right\}$ from $\{x_t\}$ once, we can perform GTF as in the standard SSM.

To include the nuisance artifacts $\{r_t\}$, one has to also swap the order of the matrix multiplication and the convolution in the full decoder model (Equation 5). This poses a problem since the $\{r_t\}$ are not convolved. Our solution is deconvolving the $\{r_t\}$ time series as well

$$\boldsymbol{J}r_t = \left\{hrf * \left(Jr^{\text{deconv}}\right)\right\}_t \tag{29}$$

$$r_s^{\text{deconv}} = \text{Conv}^{-1}(\{r_t\}, hrf)_s. \tag{30}$$

We use this as a simple mathematical trick to incorporate the artifacts into the convolution

$$x_t = \boldsymbol{B}(hrf * z)_t + \boldsymbol{J}r_t = (hrf * \boldsymbol{B}z)_t + \left(hrf * \left(\boldsymbol{J}r^{\text{deconv}}\right)\right)_t = \left(hrf * \left(\boldsymbol{B}z + \boldsymbol{J}r^{\text{deconv}}\right)\right)_t. \tag{31}$$

Finally, we obtain the relation between the deconvolved time series and the latent states as

$$x_t^{\text{deconv}} = \boldsymbol{B}z_t + \boldsymbol{J}r_t^{\text{deconv}} \tag{32}$$

$$z_t = \boldsymbol{B}^+\left(x_t^{\text{deconv}} - \boldsymbol{J}r_t^{\text{deconv}}\right). \tag{33}$$

### A.6 Additional information on Algorithm 1

In order to deal with numerical instabilities, additional hyperparameters were introduced. A too low noise level $\tilde{\sigma}$ determined by the VISUSHRINK Algorithm 2 can lead to high frequency artifacts, which can be dealt with by defining a lower noise level boundary $\tilde{\sigma}_{min}$. Although this is unlikely to occur in empirical (noisy) data, the lower noise level boundary helps to study artificial noise free data. Since the convolution treats the finite signal as periodic, artifacts at the boundaries of the computed deconvolved signal $x_t^{\text{deconv}}$ may occur. With the hyperparameters cut_l, cut_r (corresponding to start and end of signal, respectively) one can therefore define absolute cutoff times, either by integer or by floating point values (if, e.g., a cutoff time is to be defined relative to the length of the $hrf$).

### A.7 Performance measures

If the maximum Lyapunov exponent $\lambda_{\text{max}}$ of a dynamical system is larger than 0, a necessary condition for chaos, nearby trajectories will diverge exponentially. This limits the applicability of $n$-step ahead prediction errors (PEs), as conventionally used in machine learning, to evaluate model performance, as even small numerical errors will lead to exponentially growing PEs. In processes in which we can expect chaotic behavior (like neural recordings), we therefore need performance measures which are insensitive to a system's initial conditions and yet capture the most relevant dynamical properties. Here, we use two established measures [40, 47] to evaluate the DSR, the state space divergence, capturing geometric overlap of (ergodic) distributions in state space, and the power spectrum error, capturing agreement in long-term temporal properties.

On the ALN and LEMON benchmark, we computed these two measures as average over 100 trajectories, generated by perturbing the initial state with a small Gaussian noise term ($\mu = 0$, $\sigma = .01$).

### A.7.1 Prediction Error $PE$

The $n$-step prediction error is given by

$$\text{PE}(n) = \frac{1}{N(T-n)} \sum_{t=1}^{T-n} \|x_{t+n} - \hat{x}_{t+n}\|_2^2,$$ (34)

i.e. , the mean squared error between ground truth data $\{x_t\}$ and $n$-step ahead predictions of the model $\{\hat{x}_t\}$.

### A.7.2 State space divergence $D_{stsp}$

Given an observed $N$-dimensional time series $\{x_t\}$ of length $T$ and a time series $\{\hat{x}_t\}$ with the same dimension/length generated by a model, $D_{stsp}$ measures the geometrical overlap of orbits in state space [40].

For low dimensional systems, $N \leq 6$, the state space is segregated into $k^N$ bins where $k$ is the number of bins per dimension. Each bin is given an index $i$ and we count the number of times $n_i$ the time series visited bin $i$. The relative frequency of visits is then obtained by dividing by time series length $T$

$$p_i = \frac{n_i}{T}.$$ (35)

Combining these frequencies across all bins in space results in a probability distribution which approximates the state space distribution (the occupation measure) of the underlying dynamical system. The Kullback-Leibler divergence of these empirical distributions can then be computed to assess the overlap of both systems in their state space geometry,

$$D_{stsp} = \sum_{i=1}^{k^N} p_i \log\left(\frac{p_i}{q_i}\right)$$ (36)

where $p_i$ are the relative frequencies of the observed and $q_i$ of the predicted (generated) time series. Note that $D_{stsp}$ is not a metric in the mathematical sense, but a divergence that assesses the (dis-)agreement of probability distributions.

The complexity of this binning approach scales exponentially with the observation dimension $N$ and thus becomes intractable for larger $N$. To compute $D_{stsp}$ in higher-dimensional systems, [6] use Gaussian mixture models (GMMs) with centers (means) $x_t$ and diagonal covariance $\Sigma = \text{diag}(\sigma^2, \cdots, \sigma^2)$, where $\sigma$ is a hyperparameter. The GMMs along the trajectory points are given by

$$f(y) = \frac{1}{T} \sum_{t=1}^{T} \mathcal{N}(y; x_t, \Sigma).$$ (37)

Following [30], the Kullback-Leibler divergence of the two GMMs can be computed using a Monte-Carlo sampling approach

$$D_{stsp} \approx \frac{1}{K} \sum_{i=1}^{K} \log\left(\frac{f_{obs}(y_i)}{f_{gen}(y_i)}\right) = \frac{1}{K} \sum_{i=1}^{K} \log\left(\frac{1/T \sum_{t=1}^{T} \mathcal{N}(y_i; x_t; \Sigma)}{1/T \sum_{t=1}^{T} \mathcal{N}(y_i; \hat{x}_t; \Sigma)}\right)$$ (38)

where $K$ is the number of samples drawn, $f_{obs}$ is the distribution of the observed time series, and $f_{gen}$ is the distribution of the generated time series. The binning and GMM-based measures correlate strongly in low dimensions (whereas determining the correlation in high dimensions is challenging for the stated reasons).

### A.7.3 Power spectrum error $D_{PSE}$

The state space measure introduced above discards all temporal structure in the data. To include temporal information in the model evaluation, we compare the power spectra of observed and

generated time series. For each dimension $i \in \{1, \cdots, N\}$, the scalar time series $\{x_{i,t}\}$ is converted into a power spectrum density (PSD) $\{S_{i,k}\}$. The components are computed from the Fourier transform of $\{x_{i,t}\}$

$$S_{i,k} = \frac{|\widehat{x}_{i,k}|}{T} = \frac{|\mathscr{F}\{x_{i,t}\}_k|}{T}. \tag{39}$$

The agreement in power spectra is then evaluated in terms of the Hellinger Distance (HD) as suggested by [47]. Since HD is a probability measure, the computed power spectra have to be normalized by

$$\bar{S}_{i,k} = \frac{S_{i,k}}{\sum_{j=1}^{T} S_{i,j}}. \tag{40}$$

Using the power spectra of the observed time series $p_{i,k} = \bar{S}_{i,k}(\{x_{i,t}\})$ and the generated time series $q_{i,k} = \bar{S}_{i,k}(\{\hat{x}_{i,t}\})$, the HD is then assessed as

$$D_{H,i} = \sqrt{1 - \sum_{k=1}^{T} \sqrt{p_{i,k} q_{i,k}}}. \tag{41}$$

The power spectrum error $(D_{PSE})$ is computed by averaging the HDs across all $N$ dimensions of the observed system:

$$D_{PSE} = \frac{1}{N} \sum_{i=1}^{N} D_{H,i} \tag{42}$$

Before analysis, the power spectra usually have to be smoothed with a Gaussian kernel of width $\sigma$ to reduce noise. $\sigma$ can thus be considered a hyperparameter in the evaluation process and was set to $\sigma = 1$ in the current work as in [6].

## B  Details on dynamical systems benchmarks

### B.1  Lorenz63 system

The Lorenz63 introduced in [45] was designed to describe atmospheric convection based on three dynamic variables. A two-dimensional fluid layer is uniformly warmed from below and cooled from above. It is a continuous-time dynamical system given by the following set of differential equations:

$$\frac{dx_1}{dt} = \sigma(x_2 - x_1) \tag{43}$$

$$\frac{dx_2}{dt} = x_1(\rho - x_3) - x_2 \tag{44}$$

$$\frac{dx_3}{dt} = x_1 x_2 - \beta x_3 \tag{45}$$

where $x_1$ is proportional to the rate of convection, $x_2$ to the horizontal temperature variation, and $x_3$ to the vertical temperature variation. The constants $\sigma$, $\rho$ and $\beta$ are system parameters proportional to the Prandtl number, Rayleigh number, and certain physical dimensions of the fluid-layer. For chaotic behavior, $\sigma = 10$, $\rho = 28$ and $\beta = \frac{8}{3}$ are typical settings. These settings produce the so-called "butterfly attractor" characteristic of the Lorenz system. For each data set, we drew random initial conditions $\boldsymbol{x}_0 \sim \mathcal{N}(0, 1_{3\times3})$ and simulated the system using the DynamicalSystems.jl Julia package, ([15]). $10^5$ time steps were saved and used as data set (1 : 1 training vs. test split) after discarding the first $1,000$ time points to remove transients from the data. To create the benchmark data sets, these trajectories were then convolved with the $hrf_{TR}$ function. Gaussian white noise drawn from $\mathcal{N}(0, \sigma 1_{3\times3})$ was added to all data points.

Results on the Lorenz63 system are in Table 2.

Table 2: Quantitative comparison between standard SSM and convSSM on noisy Lorenz63 data ($N_{\text{converged}}$ is the number of converged models).

| Convolution | $\sigma$ | Obs. model | $N_{\text{converged}}$ | $PE_{20}$ | $D_{stsp}$ | $D_{PSE}$ |
|---|---|---|---|---|---|---|
| $hrf_{1.2}$ | 0.01 | Standard | 80 | $0.0013 \pm 0.0001$ | $0.15 \pm 0.32$ | $0.06 \pm 0.01$ |
| $hrf_{1.2}$ | 0.01 | Conv | 82 | $0.0011 \pm 0.0001$ | $0.16 \pm 0.46$ | $0.06 \pm 0.02$ |
| $hrf_{0.5}$ | 0.01 | Standard | 92 | $0.0117 \pm 0.0018$ | $0.42 \pm 0.55$ | $0.10 \pm 0.07$ |
| $hrf_{0.5}$ | 0.01 | Conv | 90 | $0.0012 \pm 0.0001$ | $0.09 \pm 0.02$ | $0.06 \pm 0.01$ |
| $hrf_{0.2}$ | 0.01 | Standard | 22 | $0.0418 \pm 0.0099$ | $2.97 \pm 0.75$ | $0.64 \pm 0.22$ |
| $hrf_{0.2}$ | 0.01 | Conv | 92 | $0.0023 \pm 0.0001$ | $0.29 \pm 0.47$ | $0.14 \pm 0.03$ |
| $hrf_{1.2}$ | 0.1 | Standard | 86 | $0.0387 \pm 0.0002$ | $0.45 \pm 0.02$ | $0.09 \pm 0.01$ |
| $hrf_{1.2}$ | 0.1 | Conv | 86 | $0.0158 \pm 0.0001$ | $0.46 \pm 0.17$ | $0.09 \pm 0.01$ |
| $hrf_{0.5}$ | 0.1 | Standard | 93 | $0.0538 \pm 0.0016$ | $0.47 \pm 0.10$ | $0.10 \pm 0.02$ |
| $hrf_{0.5}$ | 0.1 | Conv | 85 | $0.0145 \pm 0.0001$ | $0.44 \pm 0.27$ | $0.09 \pm 0.01$ |
| $hrf_{0.2}$ | 0.1 | Standard | 54 | $0.0893 \pm 0.0087$ | $2.79 \pm 0.90$ | $0.58 \pm 0.22$ |
| $hrf_{0.2}$ | 0.1 | Conv | 84 | $0.0191 \pm 0.0002$ | $0.70 \pm 0.08$ | $0.24 \pm 0.02$ |

## B.2 ALN model

The adaptive linear-nonlinear (ALN) cascade model is a population model of spiking neural networks. The dynamical variables of the ALN model describe the average firing rate and other macroscopic variables of a randomly connected, delay-coupled network of excitatory and inhibitory adaptive exponential integrate-and-fire neurons (AdEx) with non-linear synaptic currents [3].

We used *neurolib* to create neural activity generated by an ALN model [12]. In *neurolib*, the firing rate of the excitatory subpopulation of every brain area is used to simulate the BOLD signal via the Balloon–Windkessel model (for formula see [11]). As an alternative, we implemented the BOLD decoder model (Equation 5) as model linking the latent states $z_t$ (i.e., the latent neural activity corresponding to the excitatory firing rates) to the observed BOLD signal. In order to create interesting dynamics, certain values were altered from the authors' default settings, 'sigma_ou' = 0 and 'b' = 5.0. Furthermore, only the first 16 dimensions of the structural connectivity matrix 'Cmat' and the delay matrix 'Dmat' (see explanatory notebook provided by [12]) were used for comparability with the empirical LEMON data set.

*neurolib* produces simulated (latent) neural activity with a sampling rate of 0.1ms. To stay in a comparable regime with the fMRI and Lorenz time series, we chose a sampling rate of 0.5s for the simulated (observed) data. To achieve this without loss of critical information, the neuronal activity as well as the BOLD time series were decimated using a 30 point finite impulse response (FIR) filter with Hamming window. Furthermore, the neural activity time series was smoothed with a Gaussian kernel with standard deviation $\sigma = 1$ and length 5, and then standardized.

Outliers in DSR measures (Table 3) were removed using the interquartile range (IQR) method. The IQR method considers values as outliers if they are 1.5 IQRs above the third ($Q_3$) or below the first ($Q_1$) quantile (where $IQR = Q_3 - Q_1$).

In Table 3 and corresponding Figure 2D, we present the quantitative comparison between the standard SSM, the convSSM, the convSSM without GTF, and two benchmark conditions. Figure 5 depicts correlations between measures computed on different subsets of the ALN dataset.

Table 3: DSR measures evaluated on the ALN data set for the convSSM, the standard SSM, and the convSSM trained without generalized teacher forcing by setting $\alpha = 0$. Measures were evaluated on the ground truth latent space and the noisy observation space on the different created test sets.

| | | metric | ConvSSM | Standard SSM | No GTF ($\alpha = 0$) | White noise | Fixed point |
|---|---|---|---|---|---|---|---|
| observational | full pseudo-empirical time series | $D_{stsp}$ | $2.35 \pm 1.01$ | $2.16 \pm 0.86$ | $4.97 \pm 1.48$ | 3.49 | 5.15 |
| | | $D_{PSE}$ | $0.28 \pm 0.04$ | $0.3 \pm 0.04$ | $0.41 \pm 0.15$ | 0.79 | - |
| | | 10-step PE | $0.98 \pm 0.23$ | $0.54 \pm 0.13$ | $1.11 \pm 0.29$ | - | - |
| | pseudo-empirical test set | $D_{stsp}$ | $4.49 \pm 1.49$ | $3.82 \pm 1.19$ | $5.37 \pm 1.98$ | 3.68 | 4.81 |
| | | $D_{PSE}$ | $0.22 \pm 0.02$ | $0.22 \pm 0.03$ | $0.29 \pm 0.1$ | 0.76 | - |
| | | 10-step PE | $1.46 \pm 0.47$ | $1.91 \pm 0.51$ | $1.12 \pm 0.41$ | - | - |
| | Ground truth test set | $D_{stsp}$ | $2.52 \pm 1.53$ | $3.13 \pm 1.7$ | $5.19 \pm 0.59$ | 2.77 | 4.89 |
| | | $D_{PSE}$ | $0.37 \pm 0.11$ | $0.43 \pm 0.1$ | $0.54 \pm 0.23$ | 0.81 | - |
| | | 10-step PE | $1.55 \pm 0.2$ | $1.86 \pm 0.22$ | $1.2 \pm 0.19$ | - | - |
| latent | full pseudo-empirical time series | $D_{stsp}$ | $6.17 \pm 1.75$ | $7.26 \pm 1.53$ | $15.03 \pm 2.4$ | 9.84 | 15.2 |
| | | $D_{PSE}$ | $0.45 \pm 0.04$ | $0.55 \pm 0.03$ | $0.48 \pm 0.18$ | 0.58 | - |
| | | 10-step PE | $3.97 \pm 0.61$ | $3.31 \pm 0.48$ | $2.76 \pm 0.43$ | - | - |
| | pseudo-empirical test set | $D_{stsp}$ | $9.25 \pm 3.08$ | $9.98 \pm 2.99$ | $15.67 \pm 4.4$ | 11.71 | 15.27 |
| | | $D_{PSE}$ | $0.41 \pm 0.04$ | $0.5 \pm 0.05$ | $0.42 \pm 0.13$ | 0.56 | - |
| | | 10-step PE | $4.47 \pm 0.87$ | $3.7 \pm 0.75$ | $2.85 \pm 0.73$ | - | - |
| | Ground truth test set | $D_{stsp}$ | $5.86 \pm 2.13$ | $7.64 \pm 2.19$ | $14.99 \pm 0.94$ | 7.35 | 14.76 |
| | | $D_{PSE}$ | $0.5 \pm 0.08$ | $0.62 \pm 0.05$ | $0.56 \pm 0.25$ | 0.58 | - |
| | | 10-step PE | $4.41 \pm 0.47$ | $3.65 \pm 0.3$ | $2.81 \pm 0.26$ | - | - |

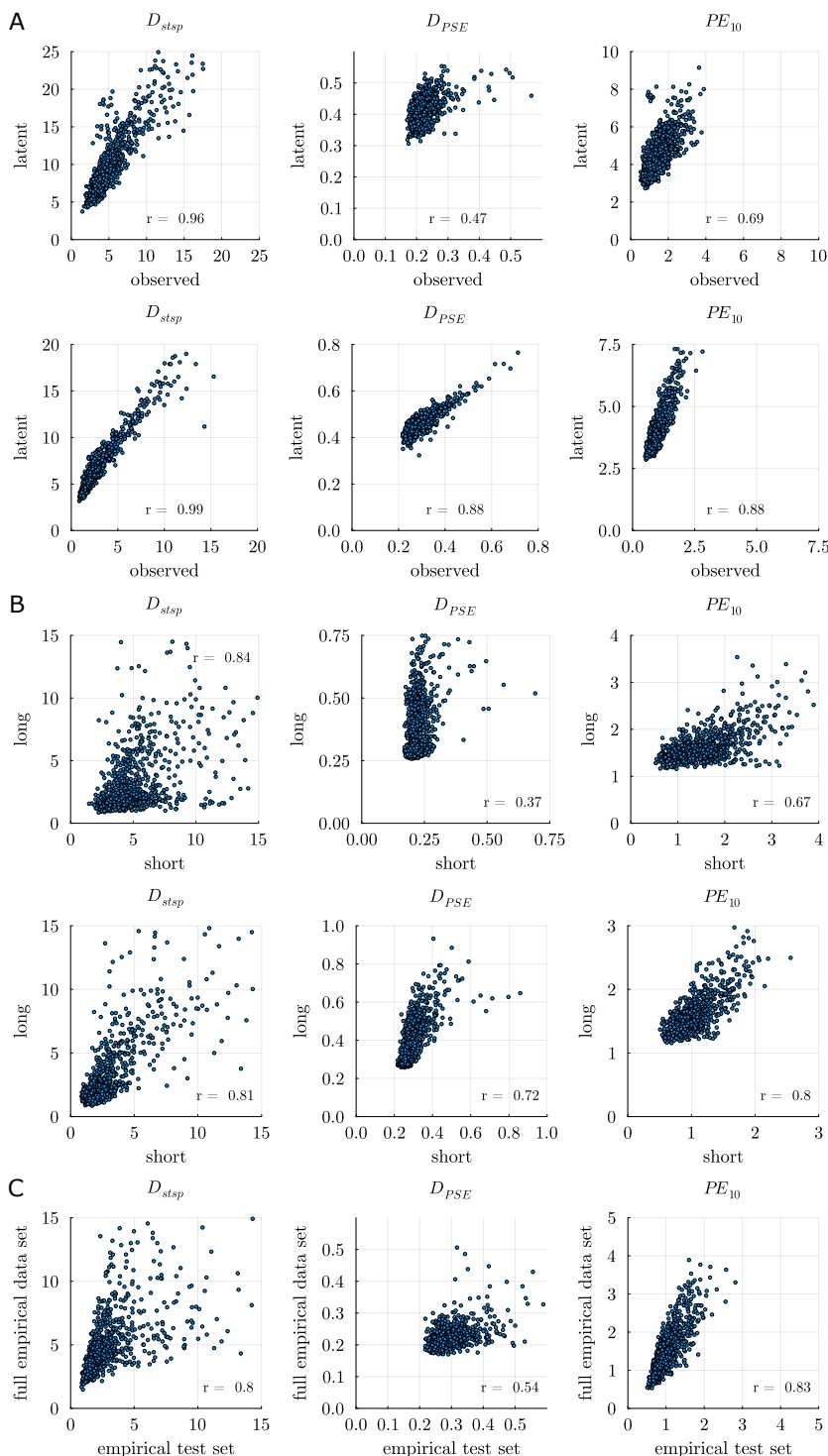

Figure 5: A: Agreement in DSR measures assessed on the observed (*x-axis*) vs. latent (*y-axis*) space of the short pseudo-empirical test set (top) and the full pseudo-empirical time series (bottom). Correlations between $D_{stsp}$ (left), $D_{PSE}$ (middle), and $PE_{10}$ (right) are displayed, respectively. B: Top: Agreement in DSR measures assessed on the pseudo-empirical test set (short) vs. GT test set (long). Bottom: Same for full pseudo-empirical time series (short) vs. GT test set (long). Correlations between $D_{stsp}$ (left), $D_{PSE}$ (middle), and $PE_{10}$ (right) are displayed, respectively. C: Correlations between DSR measures between pseudo-empirical test set and full pseudo-empirical time series, same order as in B.

For the LEMON data set, we observed non-stationarity in the data that partly resulted in performance degradation. We therefore only included participants with nearly constant variance over time. To remove non-stationary data sets, we assessed the moving average of the variance over time (window size $w = 40$ time steps). We then discarded data sets in which the variance changed with time (assessed by computing the correlation with time, with threshold set to $|r| > .16$).

The LEMON dataset can be found at `https://ftp.gwdg.de/pub/misc/MPI-Leipzig_Mind-Brain-Body-LEMON/`.

**B.4 Hyperparameter settings for the different experiments**

Table 4: Hyperparameter settings for the experiments conducted. 'Varies' means the respective hyperparameter was varied in the experiment.

| Hyperparameter | Lorenz benchmark | ALN benchmark | LEMON data set |
|---|---|---|---|
| latent_dim | 3 | 16 | 16 |
| gaussian_noise_level | 0.05 | 0.05 | 0.05 |
| optimizer | RADAM | RADAM | RADAM |
| start_lr | 0.001 | 0.001 | 0.001 |
| batch_size | 16 | 16 | 16 |
| model | shPLRNN | cshPLRNN | cshPLRNN |
| batches_per_epoch | 50 | 50 | 50 |
| observation_model | Identity | Identity | Regressor |
| lat_model_regularization | 0.0001 | 0.0 | 0.0 |
| end_lr | 1e-06 | 1e-06 | 1e-06 |
| device | cpu | cpu | cpu |
| gradient_clipping_norm | 10.0 | 10.0 | 0.0 |
| hidden_dim | 50 | 50 | Varies |
| sequence_length | 500 | 200 | 200 |
| MAR_ratio | 0.0 | 0.0 | 0.0 |
| obs_model_regularization | 0.0 | 0.0 | 0.0 |
| epochs | 1,000 | 1,000 | 1,000 |
| MAR_lambda | 0.005 | 0.0 | 0.01 |
| weak_tf_alpha | 0.1 | 0.2 | Varies |
| min_conv_noise | 1.0e-5 | 5e-06 | 1.0e-5 |
| train_test_split | 50,000 | 0.75 | 0.75 |
| TR | Varies | 0.5 | 1.4 |
| cut_l | 0 | 0.25 | 0.25 |
| cut_r | 0 | 0.5 | 0.5 |

All experiments were run on a system with a Xeon Gold 6248 CPU and 768 GB of RAM.

# C  Further Details

## C.1  Illustration of reconstruction measures

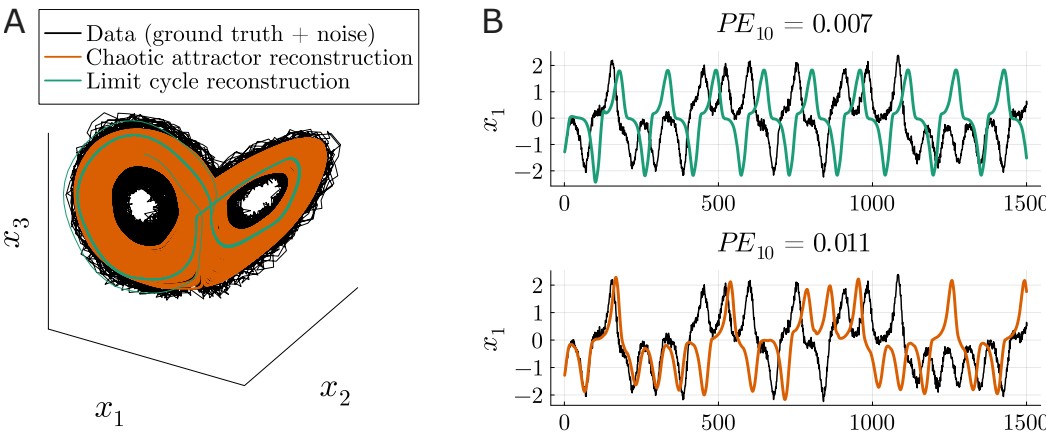

Figure 6: **A.** Ground truth Lorenz trajectory sampled with noise (black), a good reconstruction with low $D_\text{stsp}$ (orange) that accurately recovers the attractor, and a poor reconstruction with high $D_\text{stsp}$ (green) that represents the attractor inaccurately, yielding an oscillatory (limit cycle) instead of a chaotic solution. **B.** Trajectories of systems in **A.** unfolded in time. The inaccurate reconstruction (top) achieves a lower prediction error (PE) than the accurate reconstruction (bottom), due to trajectory divergence in chaotic systems. This example illustrates that PEs are inadequate to capture the reconstruction of chaotic DS.

## C.2  Comparison methods

**MINDy:** For MINDy [62] we used the implementation at `https://github.com/singhmf/MINDy`. We trained models for each subject with the settings provided by the authors for fMRI data. To obtain trajectories for calculating $PE_{10}$, $D_{PSE}$, and $D_{stsp}$, we used the provided deconvolution function to obtain an initial condition in latent space. We iterated the model forward in time for the latent trajectory and then applied the authors' observation function to output a BOLD time series which is compared with the test data.

**LFADS:** For LFADS [48], we used the lfads-torch implementation at `https://github.com/arsedler9/lfads-torch`, which provides an LFADS re-implementation in the deep learning library Pytorch [61]. The default hyperparameters provided are optimized for neural spiking data. We changed the observation model (in the framework this is referred to as reconstruction target) to a Gaussian, and changed the start and stop learning rates from $lr_{start} = 4 \cdot 10^{-3}$, $lr_{end} = 1 \cdot 10^{-5}$, to $lr_{start} = 4 \cdot 10^{-4}$, $lr_{end} = 1 \cdot 10^{-6}$, which improved the fit to our data. To obtain trajectories, we iterated the trained models forward in time with initial conditions from the test dataset using the provided *model.predict_step* function.

**rSLDS:** For rSLDS [44], we used the implementation at `https://github.com/lindermanlab/ssm`. We trained the rSLDS with the *Laplace-EM method with the Structured Mean-Field Posterior*, as recommended by the authors, with *diagonal_Gaussian* dynamics and *Gaussian_id* emissions. For a fair comparison, we used the same number of latent dimensions as for the observations (same as for the cshPLRNN). We determined the rSLDS training parameters $\alpha = 0.9$ and $K = 2$ via grid search over $\alpha \in [0.0, 0.1, 0.2, 0.3, 0.4, 0.5, 0.6, 0.7, 0.8, 0.9, 1.0]$ and $K \in [\![2, 15]\!]$ by inferring systems using a subset of the data and assessing the performance on the held-out set [6]. To generate trajectories for the model comparison, we first approximated the posterior of the test data and then sampled with *model.sample* using the approximated posteriors $x$ and $z$ and the test data as *prefix* input for the model.

## C.3  History dependence

To illustrate the difference in "history dependence" for latent time series $\{z_t\}$ and convolved counterparts $\{x_t\}$, an illustrative example was created: A latent time series $\{z_t\}$ was produced stochastically

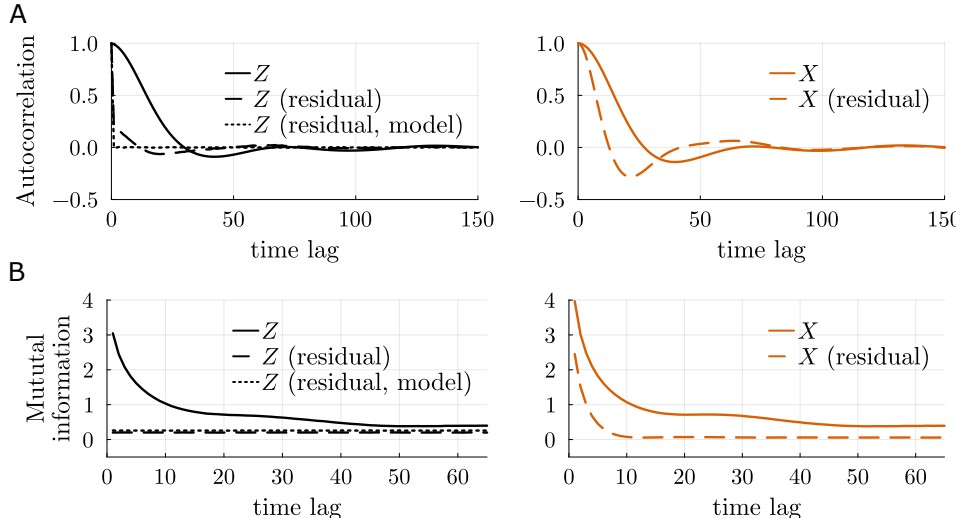

Figure 7: A: Full (solid) and residual (dashed) average (across dimensions) auto-correlation functions for the latent (left) and observed (right) time series. For the residual auto-correlation, the immediately preceding time step was regressed out. For the dotted curve, cshPLRNN($z_{t-1}$) instead of $z_{t-1}$ was regressed out. B: Same as A for the mutual information as a function of time lag.

by using a cshPLRNN equipped with a process noise term $\epsilon_t$, i.e.

$$z_t = \text{cshPLRNN}(z_{t-1}) + \epsilon_t, \ \epsilon_t \sim \mathcal{N}(0, \text{diag}(\sigma, \sigma, \sigma)).$$

The cshPLRNN employed here was trained to become a surrogate model for the Lorenz63 system, and $\epsilon_t$ is Gaussian white noise with standard deviation $\sigma = 0.1$ on each dimension. Note that there are no correlations between noise terms at different time points. A time series of length $T = 10^5$ was simulated. The corresponding observation time series $\{x_t\}$ was created by convolving $\{z_t\}$ with the hemodynamic response function $hrf_{0.5}$ for TR = 0.5 s.

We then computed auto-correlation functions for the actual and a residual time series, where for the latter the linear effect of $z_{t-1}$ on $z_t$ (and, likewise, $x_{t-1}$ on $x_t$) was removed (similar to a partial auto-correlation). As Figure 7A shows, the autocorrelation of the residual time series drops much faster, instantaneously at first, for the latent states (left) as compared to the observed/convolved variables on the right, illustrating the convolution effect is removed in the model's latent space. It is not completely gone if only linear dependencies are removed – if the model forwarded-iterated states cshPLRNN($z_{t-1}$) are regressed out instead, the auto-correlation immediately drops to zero for the latent states (dotted lines in Figure 7, left), as it should by model definition. Likewise, the (nonlinear) mutual information (Figure 7B) shows there are no temporal dependencies left in the residual latent series, while still present in the residual observed series, 'empirically' confirming our approach does what it is supposed to do.

### C.4 Acronyms

**SSM**: State space models
**DS**: Dynamical systems
**DSR**: Dynamical systems reconstruction
**TF**: Teacher forcing
**BOLD**: Blood oxygenation level dependent
**fMRI**: Functional magnetic resonance imaging
**SLDS)**: Switching Linear
**TVB**: The Virtual Brain
**DCM**: Dynamic Causal Modeling
**DL**: Deep Learning
**ODE**: Ordinary Differential Equation

**RNN**: Recurrent neural network
**rSLDS**: Recurrent SLDS
**LFADS**: Latent Factor Analysis via Dynamical Systems
**STF**: Sparse TF
**GTF**: Generalized TF
**HRF**: Hemodynamic response function
**PLRNN**: Piecewise linear RNN
**shPLRNN**: Shallow PLRNN
**cshPLRNN**: Clipped shallow PLRNN
**MSE**: Mean squared error

**SGD**: Stochastic gradient descent
**EVGP**: Exploding-and-vanishing gradients problem
**SOTA**: State of the art
**TR**: Time of repetition
**PE**: Prediction error
$D_{\text{PSE}}$: Hellinger distance/Power spectrum error
$D_{\text{stsp}}$: Kullback Leibler/ State space divergence
**ALN**: Adaptive linear-nonlinear cascade model

$\lambda_{\max}$: Maximum Lyapunov exponent **GT**: Ground truth
**LEMON**: "Leipzig Study for Mind-Body-Emotion Interactions"
**rsfMRI**: Resting state fMRI **EEG**: Electroencephalography
**LSTM**: Long Short-Term Memory
**MLP**: Multi layer perceptron
**BPTT**: Backpropagation through time

