# OpenReview forum: "A scalable generative model for dynamical system reconstruction from neuroimaging data"
_NeurIPS.cc/2024/Conference — NeurIPS 2024 poster_

### Official Review · Reviewer_qvr3 · 2024-06-14

**Soundness:** 4
**Presentation:** 3
**Contribution:** 3
**Rating:** 7
**Confidence:** 3

**Summary:**

This paper proposes a SSM-based DSR algorithm, convolution SSM model (convSSM), to recover the underlying systems. To prevent exploding gradients, the model is trained by SGD with generalized teacher forcing (GTF), for both (pseudo-)invertible and signal convolved decoders. After validating and studying the proposed method by simulations (Lorenz63 and ALN), they apply it to a experimental fMRI dataset (LEMON). Overall, the paper provides a scalable model for dynamic system reconstruction, which can be helpful for both inference and prediction task.

**Strengths:**

1. The proposed method (convSSM trained via SGD+GTF) can be efficiently scales with model size and convolution length
2. The model selection strategy based on short empirical time series are practically useful.
3. The proposed models can reliably extract key dynamical system features.

**Weaknesses:**

From the model and model inference perspective, the main contribution for the paper is training convSSM with GTF for signal convolution model, by exploiting linearity of Wiener deconvolution. The proposed method is mainly to solve the difficulty that the current observation depends on the whole series latent states, which is common in real application, especially in neuroscience.

However, the evolution of latent states described by the system equation can capture some correlation to previous states, i.e., the current latent states depends on the history of previous states. The modeling of observation equation as a function of the whole latent history further captures the remaining correlation structure, which is missed by the AR(1) assumption in system equation. But doing this makes inference harder. It would great to show the necessity of complicated modeling on observation equation, rather than put all correlation modeling into latent space (such as using GP, which is clearer and usually easier for inference), based on some evaluation criter

**Questions:**

The paper is clearly written. But the major question is the same as weakness, why it is necessary put correlation modeling into the observation equation, and hence make it more difficult to inference?

**Limitations:**

Using AR(1) for system equation and simultaneously making current ovservation explicitly depending on the whole latent history may make inference difficult.

---

> ### Author Rebuttal · Authors · 2024-08-06
>
> **Summary**
>
> We thank the reviewer for taking the time to comment on and read our manuscript in detail, as well as for this supportive and positive evaluation. We provide new results and figures in an uploaded PDF, and replies to the questions and comments below. A full list of references is listed in the general rebuttal to all reviewers.
>
>
> **Weaknesses**
>
> [“*W1*”]
>
> Thank you for your detailed review. We would like to clarify a few points regarding our model, which may not have been fully articulated in the paper. First, the AR(1) component, representing the Markov property in our latent model, is crucial because it ensures that the model constitutes a true dynamical system. If the dynamics in state space were not Markovian, it would mean the state space is not complete (lacking dimensions/ variables) and trajectories not uniquely resolved [13, 15]. For a true state space, the future must be determined solely by the current position, and the same state cannot be associated with different futures. This is where the delay embedding theorems come in in partially observed systems [16].
> An AR(1) process is therefore essential to recover the underlying dynamical system, and be able to analyze its dynamics unambiguously, such as the maximum Lyapunov exponent (e.g., Fig. R2).
>
> Second, our model is designed to reconstruct the generative dynamical system underlying neural signals measured through functional magnetic resonance imaging (fMRI). These neural signals are not directly accessible due to the delay in the delivery of nutrients and oxygen to cells, which results in a convolution with the hemodynamic response function (HRF).  To address this, we convolve the underlying neural state with the HRF, a standard practice in the fMRI research community. This approach allows us to accurately model the neural dynamics by separating them from the neurovascular response.
> By incorporating the HRF into our observation model, we disentangle the neural state and its dynamics from the neurovascular mechanics, which are not the focus of interest. We will further clarify this aspect in the updated version of our paper.
>
>
> **Questions**
>
> [“*Q1*”]
>
> The necessity of having an observation model that relates a history of states to the observed measurement stems from the actual biophysical properties of the blood-oxygenation-level-dependent (BOLD) signal measured using magnetic resonance imaging. As alluded to in response to *W1* above, when neurons in the brain are active they require oxygen. The fMRI signal measures changes in blood oxygenation in response to neural activity (the BOLD signal). To provide oxygen and nutrients, there is therefore a change in blood flow, however, it comes at a delay and with a characteristic shape, accounted for by the HRF. So the reason we have to convolve with the HRF is simply due to the biological properties that give rise to the data. We therefore agree that it makes inference more difficult, however, it is necessary to capture the underlying neural process in an unbiased manner (as now also illustrated in Figs. R1 & R2).
>
> **Limitations**
>
> [“*L1*”]
>
> As mentioned earlier, while incorporating additional lags into the latent model might simplify inference from a function approximation perspective, it would not result in a true dynamical system (see also response to *W1*). Our primary interest lies in accurately recovering the underlying system dynamics, which requires preserving the Markov property inherent to a true dynamical system. This focus allows us to explore the system's fundamental dynamics, as highlighted in [6]. Our approach ensures that we maintain the integrity of the system recovery, which is crucial for our analysis and objectives.

---

> > ### Comment · Reviewer_qvr3 · 2024-08-09
> >
> > Thanks for the authors for their detailed explanations. Most my concerns are resolved, and hence I raised my rating with 1 more score.

---

> > > ### Author Response · Authors · 2024-08-09
> > > **Response to comment**
> > >
> > > We thank the reviewer for engaging with our rebuttal, and are happy to hear we could address the open points!

---

### Official Review · Reviewer_6CX8 · 2024-07-09

**Soundness:** 4
**Presentation:** 4
**Contribution:** 3
**Rating:** 7
**Confidence:** 3

**Summary:**

The authors introduce a novel algorithm for dynamic system reconstruction (DSR) suited for systems where current observables depend on an entire history of previous states, which notably includes fMRI signals (BOLD signals) and calcium imaging, as both signal are filtered with a response function. The algorithm extends a previous class of methods (teacher forcing - TF) and employs a deconvolution pre-training step where the deconvolved signal is recovered via Wiener deconvolution, which is then used as in the standard TF paradigm. The pre-training step scales linearly with system size enabling efficient application to large datasets. Performance on synthetic dataset show better performance of the novel algorithm as opposed to standard TF.

**Strengths:**

- *Originality & Significance:* The paper presents a novel algorithm (convSSM) that extends the applicability of DSR to systems where the observables depend on the history of the latent variables. This is particularly significant as this class encompasses relevant experimental settings in neuroscience such as BOLD signals in fMRI or calcium imaging. Moreover the presented algorithm shows improved performance as compared to the previous (sota) techniques.
- *Quality & Clarity:* The paper is nicely crafted and written, with clearly organized sections. The experiments are designed with careful control to ensure meaningful comparisons.

**Weaknesses:**

- In the application of the DSR technique to the empirical LEMON dataset the comparison with the competing standard SSM technique is missing. It would have been helpful to identify whether the two techniques differed significantly in a real-world application and whether the novel algorithm offered better insights.
- Minor point: In Figure 2B/C is not clear whether the plots of $x_1$ and $x_3$ belong to panel B or not as panel C sits in between. Panel labels have generally exaggerated font size.

**Questions:**

- In Figure 2D, leftmost panel, the conv model does not seem to improve on the quality of the observables. Does this have implications for the generative-mode of the inferred models, i.e. the conv-SSM do not offer better generative performances?
- How does the standard SSM technique perform on the LEMON dataset? Does the convSSM method provide better insights?

**Limitations:**

The authors have adequately addressed the limitations of their work

---

> ### Author Rebuttal · Authors · 2024-08-06
>
> **Summary**
>
> We thank the reviewer for taking the time to comment on and read our manuscript in detail, as well as for this supportive and positive evaluation. We provide new results and figures in an uploaded PDF.
>
> **Weaknesses**
>
> [“*W1*”]
>
> We apologize for the error in the table labeling. The standard is indeed found in Table 1, where it is referred to as LinSSM (for linear SSM). We will correct this.
>
> Overall, the convSSM outperforms the standard SSM on the Lorenz63 benchmark and on the ALN data set in latent space. It performs comparably to the standard SSM on the LEMON data where the latent space is inaccessible. We therefore dissected performance contributions of the convSSM in detail by adding an additional (simple to visualize 2D) benchmark, the Van der Pol nonlinear oscillator (VdP), which we could specifically adjust to produce oscillations in a frequency range consistent with the empirical fMRI data. We show that the convSSM outperforms the standard SSM in latent space (where the VdP lives in; Fig. R1), even if we deconvolve the standard SSM in latent space. This demonstrates that the standard SSM, while a very powerful tool for dynamical system reconstruction when trained by GTF, is not able to reproduce the true underlying process without bias. We also show that this can result in biased estimates of maximum Lyapunov exponents (as now illustrated on the Lorenz system where the max. Lyapunov exponent is known, Fig. R2).
>
> In a sense, compared to the standard SSM trained with GTF, the convSSM implements a biological prior, which enables it to often capture strongly low-pass filtered (convolved) processes more efficiently, and to provide a more accurate description of the underlying latent dynamics. In that sense, yes, we believe the novel algorithm offers better mechanistic insights, as the main findings with respect to the empirical data is that we predominantly find chaotic systems.
>
> [“*W2/Minor point*”]
>
> Thank you for pointing this out. We will adjust Figure 2B/C to make it more clear, and reduce the font size in the panel labels.
>
>
> **Questions**
>
> [“*Q1*”]
>
> As pointed out in the response to *W1*, the latent dynamics is what we want to analyze in terms of computational mechanisms, and therefore care about reproducing accurately. The standard SSM trained by GTF produces biased estimates of this system (as now further illustrated in Figs. R1& R2 in the uploaded PDF). Therefore, while yes, the prediction performance is comparable, the analysis of the generative mechanisms will be flawed.
>
>
> [“*Q2*”]
>
> Please see our response to *W1*, where we address these questions in detail.

---

> > ### Author Response · Authors · 2024-08-11
> >
> > Dear reviewer,
> >
> > thank you once more for your time and effort in reviewing our work. We would kindly like to ask if our rebuttal adequately addressed your questions and concerns, and whether there are any remaining questions we can clarify.

---

> > > ### Comment · Reviewer_6CX8 · 2024-08-12
> > >
> > > I thank the authors for their detailed rebuttal and for providing additional material in such a short time frame. I fell most of my concerns are now resolved and I have raised my score (1 point) to reflect the addition of further analyses.

---

> > > > ### Author Response · Authors · 2024-08-12
> > > >
> > > > Thank you. Much appreciated.

---

### Official Review · Reviewer_9P9G · 2024-07-10

**Soundness:** 3
**Presentation:** 3
**Contribution:** 2
**Rating:** 5
**Confidence:** 3

**Summary:**

The paper proposes a teacher forcing (TF) mechanism for a latent variable model where the dynamics evolve according to a (deterministic) piecewise linear RNN model and the observations are linear projections of the latent space convolved with a filter (in the case of BOLD signals, the form of that filter is known and it’s called hemodynamic response). In the presence of nuisance parameters (such as residual motion, etc.) a linear term is added to the observation to control for their effect and isolate them from dynamics. TF proves useful for robust training of the model and does a better job in recovering long-term and topological features of the chaotic attractors. Two examples in low-d and high-d synthetic data and one example in a real fMRI dataset (called LEMON) are shown.

**Strengths:**

The literature on using control theoretic ideas for model fitting and training has been expanding. This work takes a natural step towards extending TF to latent variable models with deterministic dynamics.

The writing quality is good, which makes it easy to follow the arguments and contributions.

Most time series models focus on short-term prediction and use MSE to quantify the errors. Although I’m not fully convinced (see the questions and weaknesses) developing metrics that assess the topological properties of attractors is an interesting idea.

**Weaknesses:**

The work has several weaknesses and limitations detailed below. I’m open to discussion about any of them.

- As the title and narrative of the paper suggest, the work is mostly useful for the fMRI data where a well-established convolution filter exists. Extending this to arbitrary (and learnable) observation filters or nonlinear transformations sounds non-trivial.
- The model does not incorporate noise in the dynamics space. Given the main motivation for developing the work is to analyze real-world fMRI data the common assumption of the existence of noisy dynamics is a sensible assumption to make and lack of noise in the dynamics can limit the expressivity of the model. Some control theoretic ideas for noisy dynamics are proposed before \[2,3\], a discussion on this would be helpful to orient readers.
- I didn’t quite understand why the authors used a PLRNN in the latent space. To me, it sounds like any generic RNN should be able to take advantage of teacher forcing in the proposed framework. If this is true, please include examples with other generic RNNs (with tanh or other nonlinearities). Otherwise please explain why it’s critical to use PLRNN as the dynamics model.
- The introduction could be improved. Some relevant models are not introduced or discussed. In my understanding, some of the mean-field theory is developed not to fit the data, but to gain theoretical insight into the population dynamics. In contrast, many latent variable models are developed in the field that are not discussed in the introduction (such as nonlinear LDS, switching linear LDS and its variants, gaussian process latent variable models, data-constrained RNNs, etc.).
- Some of the results go against the main motivation and claims of the paper. Detailed comments are included below.

**Unknown observation filter**

- For the effect of _hrf_, deconvolution methods have been proposed previously. In your framework, you still run deconvolution (through using convolution in the generative model). It’s unclear whether the history-dependence still holds after controlling for the effect of _hrf_. This statistical dependency needs to be shown to motivate your work.
- _“hrf with alternative functions if we want to account for filtering in the original signal”_

Related to this, what about the cases in which the _hrf_ is not known and an a priori observation model does not exist (e.g. a neural network maps the latents to the observations)?

- In addition, the noise spectra are another unknown which is estimated using the VISUSHRIN algorithm (as discussed in the appendix). How are these choices robust to the misspecification? This robustness analysis is important given that it’s almost never the case that we can capture the true noise spectra or _hrf_.

**Results against the claims/motivation**

- _“\[40\] proved that for chaotic systems gradient-based training techniques for RNNs will inevitably lead to diverging loss gradients.”_

This is true under some assumptions, but for specific architectures, this might not hold. In order to motivate your work, it’s crucial to empirically show that the gradients diverge without TF. In fact, your results show that the model without TF does a pretty good job in learning the task and it even outperforms the models with TF on short-term performance measures. Given that the main motivation for the paper is avoiding exploding and vanishing gradients through teacher forcing, it's important to first show that it’s indeed a problem for this specific model (and datasets).

- On the LEMON dataset the latent dimension is set to be equal to the observation dimension. Some parts of the motivation of the paper come from the low dimensionality of the latent space. If the best-performing models on real data are the cases in which the latent and observation dimensions are equal, then the traditional teacher-forcing methods are equally applicable. What do the authors think about this?

**Fig. 2**

- Fig. 2E,F are labeled incorrectly?
- Unclear what’s shown in Fig. 2A.
- What are the takeaways from this figure?
- In Fig. 2C, can you include $\\lambda_{max}$ distributions from other models (specifically convNoGTF model)?

**Long-term performance measures**

- The precise definition of performance measures is in the appendix. Can you reference the appendix section in the main text for readers who want to learn more about the definitions of performance measures?
- It’s important to see how long-term and short-term performance measures are correlated. These results are in the appendix. Can the authors include references in the main text?
- In the figure, the authors show that short-term and long-term prediction errors are correlated. However, in the LEMON dataset, this trend does not seem to hold. How do the authors explain this?
- The 10-step prediction error is always better for the models without GTF. This is very surprising. First, it shows that the GTF is not necessary for this model and the models still learn the task without suffering from vanishing or exploring gradients (which goes against the main motivation of the paper).
- It looks like estimating $D_{PSE}$ in high dimensions is associated with challenges. This is perhaps confounded with dynamics and noise itself too. How trustworthy are these estimates? In other words, how much should we read into these performance measures?
- On the LEMON dataset (based on Table 1) it looks like that model without GTF achieves lower prediction error but higher $D_{PSE}$ and $D_{stsp}$. What exactly does this mean? Given the correlations reported between long and short-term performance measures don’t we expect a successful model to outperform baseline models in both measures?

**More comments**

_“This is not automatically given for standard RNN”_

No model that I’m aware of has the capability of precisely predicting neural data if there’s no clear trial structure due to non-stationarity.

_“guiding the training process through optimally chosen control signals – also referred to as teacher forcing (TF) signals”_

At least the definition that I’m familiar with does not fully coincide with this. In my understanding, there’s no notion of optimality in the control signals and they’re usually driven by a teacher model or the output feedback.

_“Chaotic systems in particular, as typically encountered in neuroscience (e.g., \[53, 21, 32\]), pose a severe problem here”_

Another paper considered the theoretical aspects of training RNNs on chaotic data \[1\], please cite and discuss it.

_“but not if it depends on a history of states”_

The assumption of history dependence is a very common assumption made in most latent variable models.

\[1\] Engelken, Rainer, Fred Wolf, and Larry F. Abbott. "Lyapunov spectra of chaotic recurrent neural networks." Physical Review Research 5.4 (2023): 043044.

\[2\] Brenner, Manuel, Georgia Koppe, and Daniel Durstewitz. "Multimodal teacher forcing for reconstructing nonlinear dynamical systems." arXiv preprint arXiv:2212.07892 (2022).

\[3\] Schimel, Marine, et al. "iLQR-VAE: control-based learning of input-driven dynamics with applications to neural data." bioRxiv (2021): 2021-10.

**Questions:**

The effect of nuisance variables is considered to be linear, why is this a good assumption?

The latent space for the Lorenz model is 3 dimensional, why do you need L=50 to model this dataset (or even larger in the next experiment)?

Comparisons with many other latent variable models are not shown. Specifically, the neuroscience community has developed a suite of latent variable models with linear, piecewise linear, or nonlinear dynamic models and linear or nonlinear observation models. How do those models compare to the datasets presented here?

**Limitations:**

See the weaknesses section.

---

> ### Author Rebuttal · Authors · 2024-08-06
>
> **Summary**
>
> Thank you for your detailed review! All refs. are found in the general rebuttal to all revs.
>
>
> **Weaknesses**
>
> [“*W1*”]
> The conv. filter could in principle also be learnable by parameterizing its length and weights. Here our focus lay on incorporating biological prior knowledge on the HRF to more accurately reconstruct the dynamics of the underlying system (Figs. R1&R2).
>
>  [“*W2*”]
> We chose BPTT as it performs competitively even for highly noisy data [2], focusing on a strong and scalable approach. However,  our framework can easily be adapted to variational inference [2-4]. The key is making the conv. filter amenable to generalized teacher forcing (GTF). Will discuss.
>
>  [“*W3*”]
> Yes, as shown in [4], the architecture for dynamical systems reconstruction (DSR) is very flexible. Other act. func. also yield good performance (eg $D_{stsp}(ReLU)= 0.29 \pm 0.47$; $D_{stsp}(tanh)=0.61 \pm 1.61 $ on Lorenz). The PLRNN was chosen as it consistently yields best performance in low dims. [5], and is math. tractable in the sense that many of its topological properties can be determined semi-analytically [6,7].
>
>  [“*W4*”]
> We will elaborate on neurosci. models like DCMs [8], SLDS [9], and gaussian process models [10], and their usages apart from DSR [6].
>
>
> **Unknown observation filter (UF)**
>
>
> [“*UF1*”]
> We are unsure we fully understand the question. The biophys. assumption is that the obs. signal is generated from the latent process via conv. By incorporating this prior, we achieve more accurate reconstructions (Figs. R1&2). Should we demonstrate that before deconv. $p(x_t|z_t) \neq p(x_t|z_t…z_{t-\tau})$, while after deconv. $p(x_t|z_t) = p(x_t|z_t…z_{t-\tau})$? We are unaware of a model-independent method to show this?
>
>  [“*UF2*”]
> See *W1*.
>
>  [“*UF3*”]
> In Fig. R6, we now show noise level inference in VISUSHRINK for time series with varying noise levels and conv. filters. We also emphasize that the Lorenz exps. (sect. 3.2) demonstrate the model's robustness in inferring GT systems using *default* VISUSHRINK settings, indicating resilience to misspecification.
>
> **Results against the claims (RC)**
>
> [“*RC1*”]
> The proof in [40] is not related to architecture, but follows from the properties of GD techniques and the chain rule (the same product series of Jacobians occurs in the def. of the max. Lyap. exponent as in the loss derivatives, causing the problem). So it is indeed quite general as long as GD-based training algos. are used. We now illustrate div. (Fig. R4A) and the problem of models that architecturally prevent div. (Fig. R4B).
>
>  [“*RC2*”]
> We assume by trad. TF the ref. means that during training the obs. are provided as inputs? GTF is designed to achieve a fine balance between obs.-inferred and forward-iterated latent states that optimally controls trajectory and gradient flows. Traditional ‘ad-hoc’ methods do not work here [5, 11]. Also, trad. TF for DSR requires a mechanism to ensure consistent interpret. of model inputs during training and runtime; directly using obs. as inputs necessitates replacing them with predicted obs., causing the latent DS to lose its Markov property due to conv. Finally, the optimal model dims. are not always of equal dim., but depend on dataset and no. of obs.
>
> **Fig. 2**
>
> Fig. 2E,F: Thanks, will be adapted.
>
> Fig. 2A: Visualizes the reconstruct. performance to provide an intuition on the applied measueres. Will clarify.
>
> Fig. 2C: Yes, Fig. R3.
>
> **Long-term performance measures (PM)**
>
> [“*PM1,2*”]
> We are happy to reference Appx. defs. and figs. in the main text.
>
>  [“*PM3*”]
> We apologize for any confusion and assume the question pertains to Fig. 5 (Appx.). We compare predictions assessed on both short and longer data to show that our PMs can be accurately evaluated on short data. This is shown only on ALN data (since LEMON data is short). Fig. 5B bottom is most relevant for empirical eval, comparing PMs on 500 vs. 5000 data points. All 3 scores correlate with $r \geq 0.72$, which we find satisfactory for assessing DSR. Will clarify.
>
>  [“*PM4*”]
> There may be a misunderstanding. Our goal with DSR is to achieve agreement in long-term behavior between recons. and true systems [6]. In chaotic systems, short-term forecasting measures like PEs *can be lower for worse reconstructions* [5-6,12], as poor models might capture mean trends or dominant osc. periods that *deviate less from the true signal on average* than accurate models that capture chaos but therefore cause exponential diverg. of trajectories [11] (Fig. R6). We now show that noGTF models frequently *inaccurately* converge to fixed points and osc. (Figs. R3&4).
>
>  [“*PM5*”]
> For $D_{PSE}$, dimensionality is irrelevant since it is performed dimension-wise. For $D_{stsp}$, efficient high-dim. approximations using GMMs are available and used here (see [5,12] for eval). Thus the main question was whether these measures are reliable on short time series.
>
>  [“*PM6*”]
> See *PM4*.
>
>
> **More comments (MC)**
>
> [“*MC1*”]
> Agreed, the point here is not precise prediction, but rather capturing long-term temporal and geometric structure [6]. Will clarify.
>
> [“*MC3*”]
> Happy to cite and discuss.
>
> [“*MC4*”]
> Yes, true, but it still constitutes a problem for DSR. A proper DS model must be Markovian by def. to ensure the uniqueness of trajectories [13].
>
> **Questions**
>
>
> [“*Q1*”]
> We followed the established practice of modeling fMRI data with linear nuisance vars. but emphasize that our approach can easily accommodate nonlinear effects.
>
>  [“*Q2*”]
> The applied RNN is piecewise linear and more pieces are needed to approx. the nonlinearities of the true eqns.
>
> [“*Q3*”]
> We specifically chose SOTA models for *DSR*. Linear lat. models cannot produce DS properties like limit cycles and chaos. Most neurosci. models focus on inferring connectivity params., with only a few addressing DSR [eg 14]. While open to comparisons, prior experiences makes us confident that these models are no match in DSR performance.

---

> > ### Author Response · Authors · 2024-08-11
> >
> > Dear reviewer,
> >
> > thank you again for your time and effort in reviewing our manuscript. We would kindly like to ask if our rebuttal adequately addressed your major concerns, or if there are any remaining questions we can clarify.
> >
> > Due to the strict character limit, our responses were partly brief. We are happy to elaborate on points that may yet be unclear.

---

> > ### Comment · Reviewer_9P9G · 2024-08-12
> > **Updated review**
> >
> > Thank you for replying to all questions and comments. In particular, the addition of new experiments and results are very helpful for a more in-depth understanding of the paper. I will slightly increase my score and would appreciate it if the authors include the following.
> >
> > **W1-3,Q1)** Can you include these two in the discussion? Specifically, can you describe in the discussion part of the paper how to extend the method to learnable filters (W1), how to extend to variational schemes for models with noise in the latent space (W2), how to extend GTF to arbitrary choices of the architecture in addition to PLRNN (W3), and how to extend to nonlinear nuisance model (Q1)? This could open up new applications of the method and make the paper more accessible to a wider audience. Would you include the updated discussion here for a review?
> >
> > **UF1)** A simple check is to look at the mutual information between x_t, x_{t-1:t-\tau} and compare it to the mutual information between z_t, z_{t-1:t-\tau}. If these two are largely different, it shows that deconvolution has successfully removed the history dependence.
> >
> > **UF3, RC1)** The new experiments address these two.
> >
> > **RC2)** Can you add some details on model selection to the discussion? Specifically, how did you set the latent dimension for specific datasets? This would be important for practitioners who’d be interested in applying this model to other datasets.
> >
> > **PM3)** Apologies for the unreferenced question. This question is mostly addressed by Fig. R3,4,6 but just to double-check; from what I understand after reading the rebuttal the authors are arguing against the use of PE (either 1-step or 10-step) as it doesn’t represent “long-term” behavior of the dynamical systems and instead suggest using D_{stsp} or D_{PSE}. Is the point of Fig. 2E to say that increasing the horizon of prediction doesn’t help with better quantification of DSR?
> >
> > **Q2)** Previous switching linear approaches have shown successful reconstruction using 2 pieces. I guess my question is why do we need the number of states to be this large for a good reconstruction?
> >
> > **Q3)** At least a comparison against one of the switching models (e.g. SLDS or rSLDS) and a nonlinear model (e.g. LFADS) would be very helpful just to make your point.

---

> > > ### Author Response · Authors · 2024-08-13
> > > **Response to updated review**
> > >
> > > W1-3,Q1) This is a very good idea, thank you for the great suggestions! We add here two paragraphs that we will integrate into the Discussion:
> > >
> > > “We emphasize that the proposed framework is highly flexible due to its modular structure, and may be easily adapted to meet diverse requirements. First, the latent model can be replaced with any other differentiable and recursive dynamical model, such as e.g. LSTMs. The GTF training framework would remain unchanged as the control signal and the latent state update (eqn. (3)) are not affected by such modifications [5]. Likewise, the observation model can easily be adapted to account for nonlinear effects of nuisance covariates, e.g. through basis expansions in these variables, or through learnable but regularized MLPs. While our model was designed as a scalable method to integrate biological prior knowledge on convolution filters like the HRF, alternatively we can parameterize the filter weights within the observation model, making them learnable through BPTT, with filter length either as a hyperparameter, or by imposing a regularization that truncates filter length by driving coefficients to zero. To prevent conflicts between filter adjustment and latent model, a viable strategy may be stage-wise learning as suggested in [12]. Once the filter is adjusted, one may reduce the learning rate on the observation model, or even fix its parameters, to prioritize learning of the dynamics. Fixing the filter parameters after an initial stage would have the advantage that subsequent training would enjoy the same speed benefits as in our suggested method.
> > >
> > > Finally, we would like to highlight that our framework could be adapted to accommodate noise in the latent process. For example, in Brenner et al. [4] the GTF procedure has been modified to work in the context of stochastic DSR models using variational inference. The key idea lies in introducing a (multimodal) variational autoencoder that takes the observed variables as input and maps them into the DSR model’s latent space, thereby generating the control signal required for GTF. In a similar fashion, we could replace the MVAE with the inversion in eqn. (10), thereby providing a TF signal to be used to steer a probabilistic latent DS model, i.e. its distributional mean, via eqn. (3), and use the reparameterization trick [17,18] for BP in the latent space. However, although probabilistic frameworks are appealing, ‘deterministic’ BPTT has previously been shown to be (at least) comparable in terms of DSR performance, even for noisy observations and/or processes [2], such that the benefits for DSR would need to be further examined.”
> > >
> > >
> > > 17) Rezende, D. J., Mohamed, S., and Wierstra, D. Stochastic Backpropagation and Approximate Inference in Deep Generative Models. In Proceedings of the 31st International Conference on Machine Learning, 2014.
> > >
> > > 18) Kingma, D. P. and Welling, M. Auto-Encoding Variational Bayes. In Proceedings of the 2nd International Conference on Learning Representations, 2014.
> > >
> > >
> > > UF1) In a deterministic DS observed within the full state space, the complete future of a trajectory is fully determined by its initial condition, so we would not expect the MI to decay to 0 with $\tau \rightarrow \infty$ (and in fact, we checked this and it does not), but we would expect that the previous state contains all information about the current state if the deconvolution worked as intended. So - in our minds - the crucial question would be whether we have $p(z_t|z_{t-1:t-\tau})=p(z_t|z_{t-1})$ in latent space, but $p(x_t|x_{t-1:t-\tau}) \neq p(x_t|x_{t-1})$ in observation space (i.e., confirming the Markov property in latent but not observation space). These high-dimensional multiple state probabilities are, however, hard to access. For the revision, we will look into different ways we can approximate them, to illustrate this property.

---

> > > > ### Author Response · Authors · 2024-08-13
> > > > **Response to updated review**
> > > >
> > > > RC2) Sure. Most hyperparameters were adopted from previous experiments (see Appx., Table 4). For the LEMON data, we additionally determined the latent dimensions (and $\alpha$) via grid search, by inferring systems using a subset of the data and assessing the performance measures ($D_{stsp}$, $D_{PSE}$) on the held-out set (see also [2]). The latent dim. of the ALN dataset was then set to match that of LEMON. For the Lorenz63, we selected a hidden dim. in which the no-filter model performed well [5], to have a strong baseline comparison. All other hidden dims. were aligned with this setting. We will provide additional details to clarify.
> > > >
> > > > PM3) Yes exactly. The PE does not assess the generative long-term statistics and, especially when the system is chaotic, will not be informative about how well the underlying attractor has been reconstructed.
> > > > The purpose of Fig. 2E, however, was to demonstrate that assessing $D_{stsp}$ on short sequences, as common in empirical studies, still provides a good indicator for the actual long-term performance, i.e. as assessed from long sequences. This is essential for using $D_{stsp}$ to evaluate reconstruction performance on relatively short empirical data.
> > > >
> > > > Q2) We apologize for misunderstanding your question. Piecewise linear models have demonstrated good reconstruction performance with far less than 50 pieces [12]. However, increasing the number of pieces enhances the model's flexibility and improves performance. How many pieces are required also depends on the complexity of the underlying attractor, and whether one is interested in capturing the full geometry (as assessed via $D_{stsp}$) or just general topological features. Since our primary goal was to compare the filter and non-filter options, ensuring that performance differences were due to the model's efficacy in well-performing regimes [5], rather than being limited by too few dimensions, we opted for higher dim. here. Further, SLDS models differ from our approach in that they allow different parameter matrices for each regime ("piece"), while here we ensure continuity in the states and the switching in Jacobians is solely regulated through the ReLU function, i.e. matrices {A, W1, W2} are constant across state space.
> > > >
> > > >
> > > > Q3) We promise to include such comparisons in our revised manuscript. We already started runs with SLDS, but since there is only 1 day left, we are skeptical we will be able to finish this by tomorrow. We note that previous experience with LFADS did not yield comparable performance, as LFADS often tended to converge to fixed points in the long run (i.e., did not capture the underlying attractor and long-term statistics).

---

> > > > > ### Comment · Reviewer_9P9G · 2024-08-13
> > > > > **Reply**
> > > > >
> > > > > The new discussion on extensions looks great, thanks for sharing it here. Regarding UF1, [1] can help pick the right estimator (they also have an accompanying code package). I think it'd be a very compelling result to show that the filter is indeed helping with the Markovian dynamics, which would strengthen the paper's motivation. It'd be great if you could rephrase what you have above in response to RC2 and Q2 and add it to the supplementary, this information could help practitioners find the right hyperparameters.
> > > > >
> > > > > I look forward to the updated manuscript with this information and comparisons against latent variable models.
> > > > >
> > > > > [1] Czyż, Paweł, et al. "Beyond normal: On the evaluation of mutual information estimators." Advances in Neural Information Processing Systems 36 (2024).

---

> > > > > > ### Author Response · Authors · 2024-08-13
> > > > > > **Response to reply**
> > > > > >
> > > > > > As the reviewer suggested, we computed the MI between the deconvolved states $z_t$ and $z_{t-1:t-\tau}$ and between observables $x_t$ and $x_{t-1:t-\tau}$ on ten time series of the Lorenz63, using the suggested package. We indeed find that the MI for the latent process is significantly lower (with a mean of $20.59 \pm 0.01$ for the MI evaluated on the states, and $26.10 \pm 0.92$ on the observables, $p<.001$). However, given that we are yet unfamiliar with this package, we do not wish to overstate these results. Also, based on the arguments we had provided in our previous response, we still believe the more appropriate test is the one on conditional independence, on which we will continue to work.
> > > > > >
> > > > > >
> > > > > >
> > > > > > We will add the information on RC2 and Q2 to the Appendix, as requested.
> > > > > >
> > > > > >
> > > > > >
> > > > > > We thank you again for your detailed engagement with our paper!

---

> > > > > > > ### Comment · Reviewer_9P9G · 2024-08-13
> > > > > > >
> > > > > > > Great, thanks for running the analysis very quickly. Indeed the conditional independence test would be more appropriate and interesting, but the current result is already consistent with what you'd like to show. The large mutual information could indeed be driven by the consecutive time points and Markov property. I wonder if this could also be used as a strategy to pick the best hyperparameters or learn the filter in case the optimal filter needs some tuning (i.e. find the filter or find the hyperparameters that result in the highest conditional independence of $p(z_t|z_{t-1}),p(z_t|z_{t-1:t-\tau})$). Anyway, I think my comments are mostly addressed and I look forward to the final version of the paper.

---

### Official Review · Reviewer_oqqk · 2024-07-11

**Soundness:** 2
**Presentation:** 2
**Contribution:** 2
**Rating:** 5
**Confidence:** 3

**Summary:**

This paper introduces two techniques, pseudo-inverse and deconvolution, for dynamical system reconstruction. The two techniques are used to help the teacher forcing for the latent sequence $z_t$ so that the learning can be more efficient. Experimental results show the effectiveness of the proposed ConvSSM + GTF compared with alternative variants.

**Strengths:**

* The two techniques seems effective for training an SSM model and dynamical system reconstruction.
* Experimental results on both synthetic and real-world datasets show that ConvSSM with GTF is better than others.
* This works might have broader impact to the field of computational neuroscience, since lots of models are based on dynamical system.

**Weaknesses:**

* From my understanding, there is actually no new model but different ways of training a particular model, although they are called ConvSSM, LinSSM, etc.
* In most recent dynamical systems, the latent RNN procedure is not a deterministic progress, but latent sequence $z_t$ is usually treated as latent variable with some noise at each time steps, such as Gaussian. It is not clear whether this work can be generalized to them.
* The presentation should be improved. It is quite hard, at least for me, to get the main idea of the model until I arrive at line 125. Lots of sentences in the abstract and introduction section seems a bit verbose and distracting.

**Questions:**

/

**Limitations:**

/

---

> ### Author Rebuttal · Authors · 2024-08-06
>
> **Summary**
>
> We thank the reviewer for taking the time to comment on and read our manuscript in detail.  We provide new results and figures in an uploaded PDF. A complete list of references can be found in the general rebuttal to all reviewers.
>
> **Weaknesses**
>
> [“*W1*”]
>
> We see our major contributions as follows:
>
> 1) We developed the first of its kind approach that makes efficient large-scale inference of dynamical systems (DS) models from empirical systems observed through convolved signals feasible. This is achieved by reformulating a SSM model such that it a) becomes amenable to recent, highly efficient control-theoretic training methods for DS reconstruction, and b) performing the deconvolution in a computationally very effective way. This leads to an algorithm which efficiently scales up to large amounts of data. In our minds this is of huge practical relevance for the field, as it enables to construct a large battery of single subject-level models in comparatively short time (and, after all, the tremendous success of some recent methods, e.g. structured SSMs like Mamba [1], lies less with the novelty of the ingredients per se, rather than with their high computational efficiency).
>
> 2) Apart from methods development, we also, for the first time, explicitly demonstrate that training and DS assessment of the models is indeed valid even on such short time series as provided by fMRI, which we think is important for the community to know. We also show that the Lyapunov spectrum can be retrieved from the trained models, something that is not possible directly on the experimental data itself.
> We believe that the novelty of our contribution should therefore be judged by how this whole framework and its validation could advance the field by opening up new possibilities, not just by how novel the underlying latent model is.
>
> [“*W2*”]
>
> The choice to use conventional (deterministic) backpropagation through time (BPTT) as training algorithm is based on previous findings that, surprisingly, in the context of DS reconstruction BPTT outperforms probabilistic training algorithms like those based on variational inference (VI; [2]), even when the data are in fact highly noisy. Nonetheless, we can easily adapt the framework presented here to work within a VI training framework, as suggested in works by [2-4]. The crucial step is to make the convolutional filter amenable to generalized teacher forcing (GTF) which we have made here.
>
> [“*W3*”]
>
> Thank you for pointing this out. We will rework the presentation to emphasize the crucial ideas behind the model.

---

> > ### Author Response · Authors · 2024-08-11
> >
> > Dear reviewer,
> >
> > thank you once more for your time and effort in reviewing our work. We would kindly like to ask if we were able to address your concerns, and whether there are any remaining questions we can clarify.

---

> > > ### Comment · Reviewer_oqqk · 2024-08-12
> > >
> > > Sorry for the late response. I think the response from authors resolved my questions. And I have no further questions. In this case, I have raised my score from 4 to 5.

---

> > > > ### Author Response · Authors · 2024-08-13
> > > >
> > > > Thank you.

---

### Author Rebuttal · Authors · 2024-08-06

**General response**

We thank all reviewers for their positive and supportive feedback, for taking the time to review our work, and for providing many helpful comments and suggestions, which we address in detail below. We have prepared a PDF file with additional material and results.

Specifically, we provide additional analyses that demonstrate once more the superiority of the introduced convSSM model over the standard SSM, both trained with generalized teacher forcing (GTF). Our results now illustrate that the convSSM produces less bias in dynamical systems reconstruction, resulting in improved performance measures (Fig. R1), as well as more accurate estimation of dynamical systems phenomena (Fig. R2).

We also explicitly show that training without GTF leads to higher gradient explosions (Fig. R4A), or alternatively, when gradient explosions are avoided through architectural adjustments, it results in more bias in model estimates due to the inability to accurately reconstruct chaotic phenomena (Fig. R4B & Fig. R3). Additionally, we illustrate the robustness of the VISHUSHRINK algorithm to misspecification (Fig. R5) and explain why prediction errors (e.g., $PE_{10}$) are not an appropriate performance measure for dynamical systems reconstruction (Fig. R6). Finally, we provide an update to Fig. 2C (Fig. R3).

We hope these additions address the reviewers' main questions and concerns. Due to the character limitations, we had to significantly condense our responses at times. We are happy to provide more detailed answers to any further questions upon request.



**References**

1) Gu & Dao. (2023). Mamba: Linear-time sequence modeling with selective state spaces. *arXiv preprint* arXiv:2312.00752

2) Brenner et al. Tractable dendritic RNNs for reconstructing nonlinear dynamical systems. In *Proc. 39th International Conference on Machine Learning* (eds. Chaudhuri, K. et al.) 2292–2320 (PMLR, 2022).

3) Kramer et al. Reconstructing nonlinear dynamical systems from multi-modal time series. In *Proc. 39th International Conference on Machine Learning* (eds Chaudhuri, K. et al.) 11613–11633 (PMLR, 2022).

4) Brenner et al. Multimodal teacher forcing for reconstructing nonlinear dynamical systems. In *Proc. 41st International Conference on Machine Learning* (2024).

5) Hess et al. Generalized teacher forcing for learning chaotic dynamics. In *Proc. 40th International Conference on Machine Learning* (eds Krause, A. et al.) 13017–13049 (PMLR, 2023).

6) Durstewitz et al. (2023). Reconstructing computational system dynamics from neural data with recurrent neural networks. *Nature Reviews Neuroscience, 24(11)*, 693-710.

7) Eisenmann et al. (2024). Bifurcations and loss jumps in RNN training. Advances in *Neural Information Processing Systems2*, 36.

8) Friston et al. (2003). Dynamic causal modelling. *Neuroimage 19*, 1273–1302

9) Ghahramani & Hinton. (2000). Variational learning for switching state-space models. *Neural Computation 12*, 831–864

10) Yu et al.  (2009). Gaussian-process factor analysis for low-dimensional single-trial analysis of neural population activity. *Journal of Neurophysiology 102*, 614–635

11) Mikhaeil et al. On the difficulty of learning chaotic dynamics with RNNs. In *Proc. 35th Conference on Neural Information Processing Systems* (eds. Koyejo, S. et al.) (Curran Associates, Inc., 2022).

12) Koppe et al. (2019). Identifying nonlinear dynamical systems via generative recurrent neural networks with applications to fMRI. *PLoS Computational Biology, 15(8)*, e1007263.

13) Perko, L (2013). Differential equations and dynamical systems (Vol. 7). Springer Science & Business Media.

14)  Singh et al.  (2020). Estimation and validation of individualized 460 dynamic brain models with resting state fmri. *Neuroimage, 221*:117046.

15) Strogatz, SH. Nonlinear Dynamics and Chaos: With Applications to Physics, Biology, Chemistry, and Engineering (CRC, 2018).

16) Sauer et al. (1991). Embedology. *Journal of Statistical Physics 65*, 579–616

---

### Decision · Program_Chairs · 2024-09-25

**Decision:**

Accept (poster)

**Comment:**

The reviewers were generally positive about the contribution, the proposed method was found well presented and sound. The reviewers ackonwledged the novelty and interest of the proposed approach to dynamical system reconstruction combining convolution state-space models with generalized teacher forcing. The experimental validation was also found appropriate. Specific questions were on the novelty and generality of the contribution, and on the deterministic nature of the dynamics considered in the latent space. It was also recommended to provide a comparison with respect to models based on simpler assumptions.

The rebuttal provided further experimental assessment on the benefit of the proposed approach over a range of benchmarks (e.g. to assess the advantages of training with GTF as compared to SGD). The authors provided further arguments to position the contributions with respect to the state-of-the-art, in particular with respect to models implementing stochastic latent dynamics. The discussion phase was rich of interactions and allowed the authors to further clarify the contributions and provide additional experimental material. Notably, the discussion identified further investigation directions, for instance relating the Markov properties in latent and observations spaces, which were found of substantial interest.

As a result, several reviewers raised the score to recommend acceptance to this work. Overall, this paper makes interesting contributions to the relevant domain of state-space modeling, and the proposed approach may have impact beyond the neuroimaging domain. For all these reason, the paper is accepted for presentation to NeurIPS 2024.